# MRONJ Treatment Strategies: A Systematic Review and Two Case Reports

Angelo Michele Inchingolo †, Giuseppina Malcangi †, Irene Ferrara †, Assunta Patano *, Fabio Viapiano, Anna Netti, Daniela Azzollini, Anna Maria Ciocia, Elisabetta de Ruvo, Merigrazia Campanelli, Pasquale Avantario, Antonio Mancini, Francesco Inchingolo *, Ciro Gargiulo Isacco, Alberto Corriero, Alessio Danilo Inchingolo ‡ and Gianna Dipalma ‡

Department of Interdisciplinary Medicine, School of Medicine, University of Bari "Aldo Moro", 70124 Bari, Italy
* Correspondence: assuntapatano@gmail.com (A.P.); francesco.inchingolo@uniba.it (F.I.);
  Tel.: +39-3472197761 (A.P.); +39-3312111104 (F.I.)
† These authors contributed equally to this work as first authors.
‡ These authors contributed equally to this work as the last authors.

**Abstract:** MRONJ is a serious drug-related side effect that is most common in people using antiresorptive and/or angiogenic medications. Therapy options for this condition include conservative treatments, surgical procedures with varied degrees of invasiveness, and adjuvant therapies. The aim of the present study is to identify the most successful and promising therapy alternatives available to clinicians. PubMed, Cochrane, Scopus, Web of Science, and Embase were searched for works on our topic published between 8 January 2006 and 8 January 2023. The search was restricted to randomized clinical trials, retrospective studies, clinical studies, and case series involving human subjects with at least five cases and no age restriction on participants. A total of 2657 was found. After the selection process, the review included 32 publications for qualitative analysis. Although conservative treatments (pharmacological, laser, and minimally invasive surgery) are effective in the early stages of MRONJs or as a supplement to traditional surgical resection therapy, most studies emphasize the importance of surgical treatment for the resolution or downstaging of advanced lesions. Fluorescence-guided surgery, PRP, PRF, CGF, piezosurgery, VEGF, hyaluronic acid, and ozone therapy all show significant potential for improving treatment outcomes.

**Keywords:** MRONJ; osteonecrosis; denosumab; DRONJ; concentrated growth factor; platelet-rich plasma; platelet-rich fibrin; bone resection; oral surgery; BRONJ

## 1. Introduction

Medication-related osteonecrosis (MRONJ) represents an uncommon disease with a frequency of about 1% (range 2–6.7%) [1]. As early as 2003, it appeared in the dental literature, and since then, many position papers have been published on this topic [2]. MRONJ is diagnosed in the case of current or previous treatment with antiresorptive drugs, exposed bone in the maxillofacial region that persists for more than eight weeks, and no history of radiation therapy to the jaws or metastatic disease of the jaws [3]. Pharmaceutical companies have warned patients about their effects on the teeth/jaw. Most dentists in their lifetime have seen one or more cases of osteonecrosis from bisphosphonates. In 2014, the World Health Organization came to recognize, after many different acronyms, "MRONJ: Medication-related osteonecrosis of the jaws", the necrotic bone in either jaw as osteonecrosis caused by some drugs [4].

Thus, the definition is "drug-related adverse reaction characterized by progressive destruction and necrosis of the mandibular and/or maxillary bone of individuals exposed to treatment with drugs for which an increased risk of disease is established, in the absence of prior radiation treatment" [5].

### 1.1. Pathogenesis

The pathogenetic mechanism of MRONJ works on the action of drugs that are active in remodeling turnover and vascularization of bone. The osteoblasts, which generate the matrix, get stocked in the mineralized matrix and transform into osteocytes with a survival of about 180 days [6]. They produce osteoprotegerin (OPG), which is a protein that inhibits the ligand RANK (reactive activator of nuclear κB ligand) and activates osteoclasts, preventing bone resorption [7].

OPG production stops with apoptosis or necrosis of the osteocyte. As a result, the RANK ligand stimulates its receptor on the osteoclast, and resorption of dysfunctional or necrotic bone tissue occurs. This mechanism maintains bone homeostasis, giving the skeleton proper elasticity and load-bearing capacity [8]. Bone in the maxillae and particularly in the alveolar processes undergoes a turnover 10 times faster than that in the long bones. This difference may explain why MRONJ almost exclusively affects this area. The most involved areas of the maxillaries are the alveolar process and, in particular, the postextraction alveoli, the mandibular posterolingual areas, the floor of the maxillary sinus, and the torus. In summary, MRONJ is mainly localized at the level of the maxillae due to peculiar features:

1. More rapid bone turnover than in the long bones;
2. Terminal-type vascularization;
3. Mucoperiosteal lining overlying the bone tissue is easily subject to trauma;
4. Presence of microflora/biofilm in the oral cavity;
5. Presence of the periodontal ligament, which in the case of dental-periodontal injury results in exposure of the underlying bone tissue [7–9] (Figure 1).

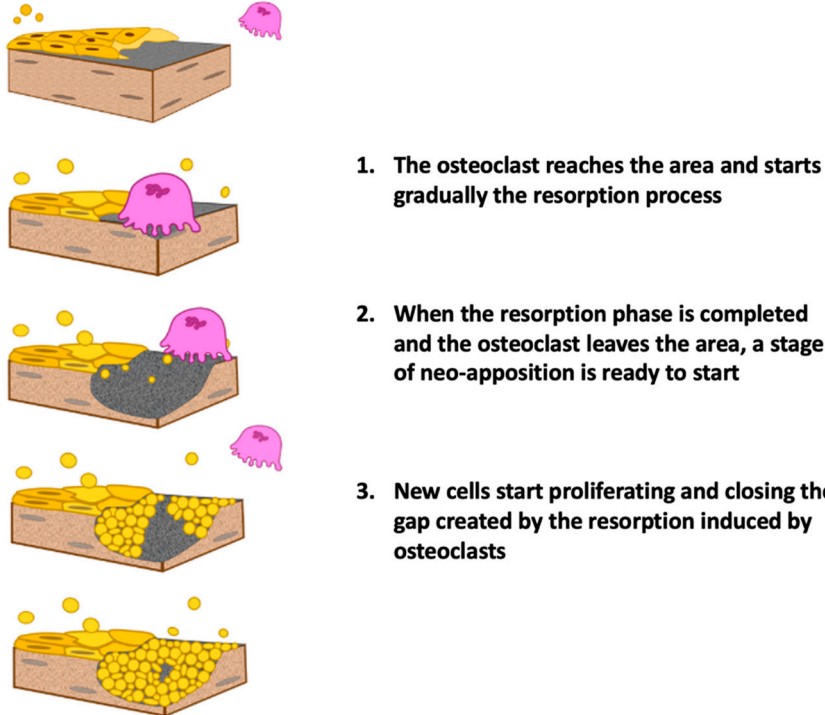

1. The osteoclast reaches the area and starts gradually the resorption process

2. When the resorption phase is completed and the osteoclast leaves the area, a stage of neo-apposition is ready to start

3. New cells start proliferating and closing the gap created by the resorption induced by osteoclasts

**Figure 1.** Role of the bone remodeling unit (BMU) in bone remodeling process.

### 1.2. Most Associated Drugs with MRONJ

The level of risk is due to the type of drug, the dose, the frequency of taking, the duration of taking, the mechanism of action, and the date of the last dose. Bisphosphonates (BPs) are analogs of pyrophosphates and strongly bind the mineral component of bone: hydroxyapatite. BPs consist of two phosphorus chains linked to a central ring consisting of a carbon atom linked to two chains R1 and R2, the former responsible for the drug's affinity to hydroxyapatite, and the latter responsible for its potency [10] (Figures 2 and 3).

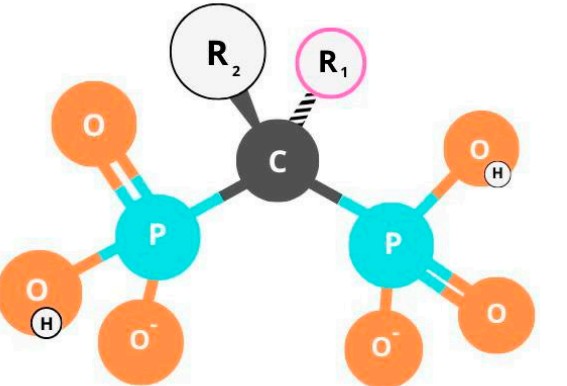

**Figure 2.** Chemical formula of pyrophosphate to BPs.

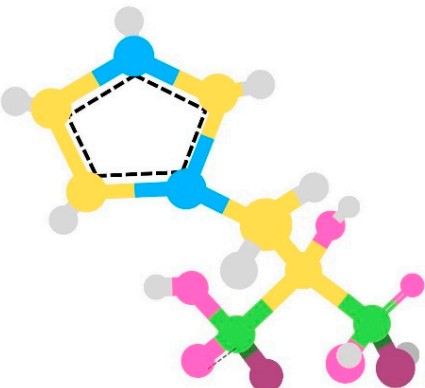

**Figure 3.** Chemical structure of BPs.

Based on the presence/absence in the R2 chain of an amine group, BPs are distinguished into two pharmacological classes: aminobisphosphonates and non-aminobisphosphonates (NBPs). The former includes zoledronate, pamidronate, alendronate, risedronate, ibandronate, and neridronate, and the NBPs clodronate, tiludronate, and etidronate. NBPs have a higher affinity for bone and 10- to 1000-fold greater potency than BPs containing no amine groups [11]. NBPs are, to date, the major category among BPs for which an association with the development of BRONJ has been identified [12] (Figure 4).

**Figure 4.** Chemical structure of the molecule of zoledronic acid (zoledronate).

In addition to BPs, another drug is widely associated with BPs: denosumab (DB). DB inhibits the RANK ligand (RANKL), which is not only required to stimulate the adult osteoclast to resorb bone but is also required at almost all stages of osteoclast maturation, starting with the mononuclear cell, i.e., the bone marrow precursor of multinucleated osteoclasts. Alendronate and DB are responsible for more than 97% of MRONJ cases in non-oncology patients treated for osteopenia/osteoporosis [13] (Figure 5).

**Figure 5.** Effect of DB and BP on bone resorption.

Other drugs have been associated with MRONJ in more recent studies:

- Tyrosine kinase inhibitors (TKIs) such as sunitinib;
- Additional monoclonal antibodies, angiogenesis inhibitors, such as bevacizumab;
- Fusion proteins such as aflibercept;
- mTOR inhibitors such as everolimus;
- Radiopharmaceuticals such as radium-223;
- Estrogen inhibitors such as raloxifene;
- Immunomodulators (methotrexate and corticosteroids) [14].

### 1.3. Half-Life

With a half-life of 11.2 years, all BPs connect severely to the bone's mineralized matrix. BPs have such a strong affinity for the bone that after the apoptosis of an osteoclast is exposed to BP, it releases BP into the extracellular matrix. The BP molecules are then quickly integrated into the neighboring bone. This cumulative mechanism of BP molecules in the alveolar bone (which possesses the characteristic of rapid turnover) could be considered the pathogenetic reason why ONJ is a disease involving the jaw bones [15].

On the other hand, DB has a short half-life (26 days) and does not bind bone, but it is so potent that it causes MRONJ, affecting osteoclast growth in the bone marrow.

### 1.4. Mechanism of Action

All BPs are cytotoxic, suppressing the cytoplasmic enzyme farnesyl synthetase. The most affected cells are osteoclasts, which absorb a large concentration of BPs and cause bone resorption during remodeling [16] (Figure 6). The toxicity of BPs is due especially to their half-life in bone and their irreversible binding to the bone mineral matrix. DB in MRONJ causes inhibition of the RANK ligand, which is necessary not only to stimulate adult osteoclasts to resorb bone but also at almost all stages of osteoclast maturation from the bone marrow precursor of osteoclasts. This potent effect makes it an important

risk factor for MRONJ, especially when administered at high doses (120 mg/month) in cancer patients [17].

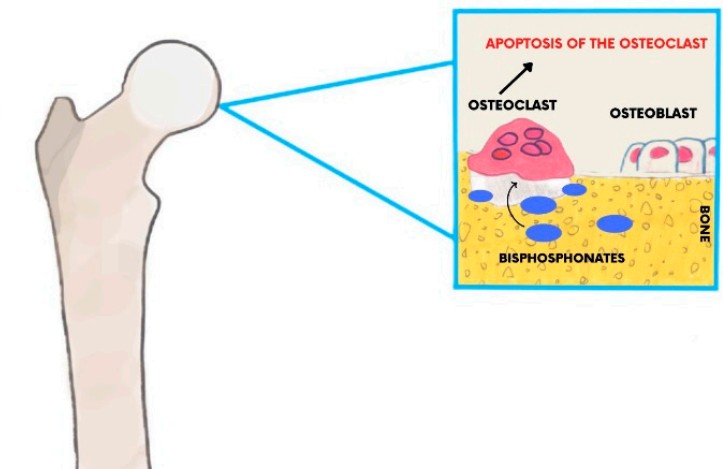

**Figure 6.** Mechanism of action of BPs on the bone.

### 1.5. Risk Factors Associated with MRONJ

The diseases most commonly associated with MRONJ are oncological diseases, osteoporotic diseases, and osteometabolic diseases. Cancer patients, in fact, take DB at high doses. Therefore, the bone contains a higher concentration of BP. Surgical procedures in cancer patients are more invasive and resective because of the high rate of recurrence [18]. Osteoporotic patients take drugs intravenously, which allows 140 times faster accumulation than the oral route. When drugs are taken subcutaneously, the risk is related to the dose and frequency of administration [19].

### 1.6. Local Causal Factors

The causative factor most associated with MRONJ is tooth extraction (around 61%). In a patient at risk of MRONJ, dental extraction determines a request for a bone replacement that the alveolar bone cannot meet. Sometimes, dental extractions can occur in a nonexposed necrotic bone. Other times, the bone is exposed by a fistula or through the perforation of a molar before extraction. We can therefore say that for more than half of the cases, the extraction can determine the identification of the MRONJ; for the remaining cases, the trauma caused by the extraction caused the injury or was already present MRONJ but not visible [20].

About 30% of MRONJ occur in the posterolingual area of the mandible, exactly where the masticatory forces are directed; it often show signs of usury and bruxism.

Chronic inflammation caused by untreated periodontitis is a factor that can cause MRONJ. Inflammation increases osteoclastic-mediated alveolar bone turnover, resulting in osteonecrosis if the patient takes BP or DB [18].

Bone biopsies, clinical crown elongation, bone surgery, and bone insertion of a dental implant can also cause MRONJ, resulting in bone trauma; therefore, there is a need for bone turnover, which BPs or DBs inhibit.

### 1.7. Staging

Currently, several classifications are proposed by various authors, which should help the surgeon conduct the correct treatment plan according to the severity of the clinical picture, the cost-benefit ratio, and the patient's various comorbidities.

Favia et al. in 2014 proposed a dimensional staging of MRONJ to evaluate the treatment strategy accurately:

- Stage 0—Clinical symptoms and nonspecific radiological signs without exposure of the bone;

- Stage I—Exposed bone <2 cm with or without pain;
- Stage II—Exposed bone between 2 and 4 cm with pain responsive to nonsteroidal anti-inflammatory drugs (NSAIDs);
- Stage III—Exposed bone >4 cm with pain not responsive to NSAIDs + complications (fistulae, maxillary sinus, or inferior alveolar nerve involvement) [21].

### 1.8. Actual Practice Guideline

Of all the guidelines proposed by the various scientific societies, the most used by practitioners are the American Association of Oral and Maxillofacial Surgeons (AAOMS) practice guidelines drafted by Ruggiero et al. [14].

Taking a brief history from 2007 to the present, the AAOMS guidelines have undergone several modifications. In 2007 the first definition of BRONJ appeared, i.e., "BPs necrosis" with almost no risk with oral intake of BPs, while a higher risk (up to 12%) with intravenous intake. The staging ranges from zero to three: risk category; stage I—exposed bone without infection and pain; stage II—exposed bone with infection and pain with or without purulent drainage; stage III—exposed bone with infection and complications [22]. In 2009, the definition of BRONJ was increased to the following: "BPs necrosis with exposed bone >2 months with a history of radiation therapy to the jaws [23]. In 2014, MRONJ i.e., necrosis by Medicaments (BPs + DBs + antiangiogenics + TKIs), with exposed or probable bone via intra- or extraoral fistula without a history of radiotherapy to the jaws". In addition, the staging is modified with the risk category as a separate category, then stage 0: specific symptoms without exposed bone, while the rest of the stages remain the same [24].

In 2022, MRONJ-associated drugs increased (+mTOR inhibitors + radiopharmaceuticals + estrogen inhibitors + immunosuppressants). Treatment previously placed more emphasis on surgical treatment. Today, however, it is recommended in the first step to choose a more medical, nonoperative treatment [14].

This study aimed to conduct a systematic review of the literature to analyze the treatment options available for MRONJ, examining the effectiveness of each treatment.

## 2. Materials and Methods

### 2.1. Protocol

The Preferred Reporting Items for Systematic Reviews and Meta-Analyses (PRISMA) guidelines were used in this systematic review [25]. The review protocol was registered at PROSPERO under the unique number 404028.

### 2.2. Data Sources and Search Strategy

The qualifying criteria were developed using the PICOS (population, intervention, comparison, outcomes, and study design) framework. PubMed, Cochrane, Scopus, and Web of Science databases were searched from 8 January 2006 up to 8 January 2023, using the keywords "denosumab OR bevacizumab OR adalimumab OR romosozumab) AND osteonecrosis AND (treatment OR therapy) AND jaw" (Table 1). The authors checked the titles and complete texts of any papers that might be relevant.

**Table 1.** Database search indicator.

| | |
|---|---|
| Articles screening strategy | KEYWORDS: (denosumab OR bevacizumab OR adalimumab OR romosozumab) AND osteonecrosis AND (treatment OR therapy) AND jaw Boolean Indicators: ("A" AND "B") Timespan: from 8 January 2006 to 8 January 2023 Electronic Database: PubMed, Cochrane, Scopus, Web of Science, and Embase |

## 2.3. Inclusion and Exclusion Criteria

This research studies the treatment strategies for monoclonal antibody drug-related osteonecrosis of the jaw. Articles that met several criteria were included: (1) the study design selected was randomized clinical trials (RCT), case series with more than 5 case reports, clinical trials, and retrospective studies (R); (2) participants were human of any age; (3) patient affected by osteonecrosis of the jaw induced by the assumption of monoclonal antibody drugs (DB, bevacizumab, adalimumab, and romosozumab); (4) the language selected was English; (5) only full-text.

The excluded studies were characterized by one of the following exclusion criteria: reviews, letters, or comments; animal models or dry skull studies; case series with less than 5 case reports; case reports; in vivo and in vitro studies.

## 2.4. Study Selection and Characteristics

The study data was selected by analyzing the study design, number of patients, MRONJ stage of disease, type of primary disease, the average age of the sample, type of antiresorptive drugs, duration of therapy, description of treatment, follow-up, and outcomes (Table 2).

**Table 2.** Descriptive summary of item selection.

| Authors (Year) | Study Design | MRONJ-Stage (n. Patients) | N. Total Patients (n. Lesions) and Type of Primary Disease (n. Patients) | Average Age (Years) | Types of Antiresorptive Drugs (n. Patients) and Duration of Therapy (Months, Years) | Description of Treatment (n. Patients) | Follow-Up (Months) | Outcomes |
|---|---|---|---|---|---|---|---|---|
| Hayashida, S. (2017) [26] | R | I and II (356) III (71) | 427 Neoplasms (236) Osteoporosi (191) | CT: 72.7 ± 10.7 ST: 73.2 ± 10.3 | DB (107) BP (320) for 28.8 months | CT (236): antiseptic mouth rinse, systemic antimicrobial agent, or debridement. ST (191): conservative surgery, removal of only the necrotic bone; extensive surgery, removal of the necrotic and surrounding bone; marginal mandibulectomy or partial maxillectomy. | 15 months | Better outcome in low-dose antiresorptive agent and ST. Better outcome of extensive ST surgery vs. CT. According to multivariate analysis, 4 variables are significantly correlated with treatment outcome: high-dose administration of an antiresorptive agent, serum albumin level, discontinuing drug, and surgical treatment. |
| Schiodt. et al. (2017) [27] | P | I (86) II (191) III (34) N.R. (9) | 327 Neoplasms (327) | 67 | DB (63) for 15.7 months DB + BP (69) for 40 months BP (196) for 26.2 months | ST (102, 31%): debridement, sequestromy, resection with or without jaw reconstruction, curettage, and teeth extraction. CT (300, 92%): antibiotics, oral rinses, analgesics, and antifungal/antimycotic. | 11 months | Patients exposed to both DB and BP did not present with higher proportions of stage III ONJ or severe symptoms. Rates of associated local oral risk factors (such as tooth extractions) were similar in patients with combined exposure to DB and BP vs. those exposed to single agent antiresorptive. Patients that discontinued antiresorptive treatment between baseline and 3 months had a higher percentage of stage III MRONJ |
| Watanabe (2021) [28] | R | II (206) III (46) | 252 Neoplasms (133) Osteoporosis (119) | 71.2 ± 9.9 | DB (46) for 81.7 months BP (206) for 18.3 months | 138 (54.8%) underwent surgery, and 143 (56.7%) received HBO therapy. | ≥3 months | In the univariable analysis, therapeutic effect of surgery in both stage II and III vs. therapeutic effect of HBO in stage II. In the multivariable analysis for stage II, stronger association with healing of extensive surgery vs. conservative surgery. ≥46 sessions of HBO therapy less associated with healing vs. non-HBO therapy. |

**Table 2.** *Cont.*

| Authors (Year) | Study Design | MRONJ-Stage (n. Patients) | N. Total Patients (n. Lesions) and Type of Primary Disease (n. Patients) | Average Age (Years) | Types of Antiresorptive Drugs (n. Patients) and Duration of Therapy (Months, Years) | Description of Treatment (n. Patients) | Follow-Up (Months) | Outcomes |
|---|---|---|---|---|---|---|---|---|
| Favia (2016) [21] | R | I: 23<br>Ii: 187<br>III: 101 | 244 (322)<br>Neoplasms (172)<br>Osteoporotic (72) | N.R. | DB (13)<br>DB + BP (3)<br>BP (228) | ST based on the dimensional staging Stage I: surgical debridement; stage II: small open-access surgery with piezosurgery of bone margins; stage III: wide open-access surgery with extensive bone resection and piezosurgery of bone margins. | 16 months | Dmab-MRONJ is in stage II or III, requiring a more or less invasive surgical approach. Complete clinical and radiological healing in 86.9% of lesions, recurrence in 13.1% recurred (in patients who could not interrupt chemotherapy, steroids, and/or antiresorptive drugs). |
| Kaibuchi (2021) [29] | R | I (26)<br>II (76)<br>III (27) | 129<br>Neoplasm (72)<br>Osteoporosis (57) | 74.3 ±11.5 | DB (40)<br>BP (89) | CT (71): mouth rinse (saline or povidone-iodine) and systemic antibiotics CT + ST (58): removal of sequestrum | 60 months | Cure rate higher in patients with osteoporosis vs. in those with cancer and in patients who had separation of sequestrum vs. in those who did not. |
| Yoshida T. et al. (2021) [30] | R | I (13)<br>II (91)<br>III (19) | 123 osteoporotis:<br>(123) | 75.1 ± 8.4 | DB (11) for 8.9 months<br>BP (112)<br>for 91.1 months | ST: 82 (66.7%)<br>CT: 41 (33%) | >3 months | ARONJ is frequently located in the mandible (55.3%), and the most frequent initiating event is periodontitis (42.3%). The most common type of ARONJ is BRONJ (91.1%). Most patients continued the ARA therapy (80.5%). No significant difference in the healing rates between the continued group and discontinued group. |
| Favia (2018) [31] | R | I (11 lesions)<br>II (65 lesions)<br>III (55 lesions) | 106 (131)<br>Neoplasm (95)<br>Osteporosis (36) | ST: 70.2<br>CT: 71 | DB (20)<br>DB + BP (7)<br>BP (103) | ST (85 patients, 107 lesions): perioperative antibiotics, surgical removal of the necrotic bone, from simple surgical debridement for smaller MRONJ to extensive bone resection for larger lesions. CT (21 patients, 24 lesions): chlorhexidine, periodic dental checks, systemic antibiotic, monthly low-level laser therapy, and removal of bone sequestrum. | 12 months | Complete healing in all the surgically treated lesions showed, except for 13.5% of the lesions (from stage III to stage I). Stabilization without complete healing in all the lesions with conservative treatment |

**Table 2.** *Cont.*

| Authors (Year) | Study Design | MRONJ-Stage (n. Patients) | N. Total Patients (n. Lesions) and Type of Primary Disease (n. Patients) | Average Age (Years) | Types of Antiresorptive Drugs (n. Patients) and Duration of Therapy (Months, Years) | Description of Treatment (n. Patients) | Follow-Up (Months) | Outcomes |
|---|---|---|---|---|---|---|---|---|
| Hadaya (2018) [32] | R | I (47) II (32) III (4) | 106 Neoplasm (89) Osteoporosis (34) | 71.7 | DB (17) BP (98) for 5.5, 3.7, and 4.8 years | Local wound care: mechanical vigorous debridement and cleaning of exposed bone, oral antibacterial solution (chlorhexidine), and antibiotics. | 24 months | Complete disease resolution in 71% of lesions, disease improvement in 22%. Association of wound care score with disease resolution and time to resolution. No effect on resolution of demographics, anatomic site, condition, and type and time of antiresorptive treatment. |
| Wei et al. (2022) [33] | R | I (29) II and III (87) | 95 (122) Prostatic cancer (95) | 75.17 ± 8.49 | DB (42) DB + BP (17) BP (57) for 18 months | Stage I: conservative treatment with chlorhexidine mouth rinse. Stage II: chlorhexidine mouth rinse, Antibiotics, and/or analgesics. Stage III: superficial debridement, sequestrectomy, or bone resection. | 12 months | Cumulative response rate of patients treated with DB is 70.5%. DB, pretreatment level of CTX > 150 pg/mL, and anemia are independent prognostic factors of MRONJ in a multivariate analysis. |
| Osaka (2021) [34] | R | I (15) II (51) III (22) | 88 Neoplasms (15) Osteoporosis (73) | 80.5 ± 6.7 | DB: 28 BP: 60 (for 49.1± 33.4) | CT: antimicrobial mouthwash, systemic antimicrobials, or debridement of bony sequestra. ST: Conservative surgery: removal of necrotic bone area. Extensive surgery: removal of the necrotic and surrounding bone, marginal mandibulectomy, or partial maxillectomy. | 23 months | According to univariate analysis, 3 variables significantly correlated with prognoses: sex, dosage, and the treatment method. According to multivariate analysis, 2 variables significantly correlated with the prognosis: high-dose administration of an antiresorptive agent. |
| Ristow (2018) [35] | R | I (31) II (26) III (11) | 87 (104) Neoplasms (79) Osteoporosis (8) | 66.25± 9.58 | DB (33) DB + BP (12) BP (42) for 50.37 ± 32.55 months | Mylohyoideus muscle flap (MMF) for the lower jaw: 57 patients Pedicled buccal fat flap (BFF) for the upper jaw; 30 patients | 8 months | Mucosal integrity in 88.0% (44/50) of patients in the MMF group and 93.1% (27/29) of patients in the BFF group. No serious side effects were reported. Better outcome of stage I and II vs. stage III. |
| Coropciuc (2017) [36] | R | I (57) II (47) III (5) | 79 (109) Neoplasms (79) | 70 | DB (43) BP (36) | ST: minimally invasive approaches (sequestrectomy, debridement of soft tissue, and application of leucocyte and PRF) CT: oral rinses and antibiotics | ≤24 months | Complete healing and resolution of disease in 38/57 stage I lesions, 30/47 stage II lesions, and 3/5 stage III lesions. Improvement of symptoms in 16/47 stage II lesions and 2/5 stage III lesions. Fifteen of the stage I lesions and one of the stage II lesions failed to respond. |

**Table 2.** *Cont.*

| Authors (Year) | Study Design | MRONJ-Stage (n. Patients) | N. Total Patients (n. Lesions) and Type of Primary Disease (n. Patients) | Average Age (Years) | Types of Antiresorptive Drugs (n. Patients) and Duration of Therapy (Months, Years) | Description of Treatment (n. Patients) | Follow-Up (Months) | Outcomes |
|---|---|---|---|---|---|---|---|---|
| El-Rabbany (2019) [37] | R | I (15) II (46) III (17) | 78 Neoplasms (12) ( Osteoporosi (66) | 77.2 | DB (17) for 5 years BP (61) for 5 years | ST (56): debridement, curettage, sequestrectomy, cauterization, and resection. CT (22): local and/or systemic antimicrobial therapy, hyperbaric oxygen therapy, pentoxifylline, and teriparatide. | ST: 15.5 months CT: 11 months | Disease resolution in 39 ST patients (70%) vs. 8 CT patients (36%). ST was associated with disease resolution vs. CT alone, after adjustment for age, duration of antiresorptive or antiangiogenic therapy, whether the antiresorptive or antiangiogenic agents were used for oncologic purposes, and the stage of MRONJ at initial presentation. |
| Sánchez-Gallego Albertos (2021) [38] | R | II (4) II (52) III (14) | 70 Neoplasms (36) Osteoporosis (34) | 50–70 | DB (10) BP (60) for 6–12 months | Surgical treatment: resection of necrotic bone, sequestrae and refreshing bone margins with a drill, extraction of the teeth if they were near the necrotic bone, and PRP placement. | 2–52 months | More recurrence (18.6%) in breast cancer patients, smokers, and patients that had been administered zoledronic acid. Smoke is the only independent risk factor. |
| Akashi (2018) [39] | R | I (17) II (28) III (15) | 61 MRONJ Neoplasms (28) Osteoporosis (33) 27 ORN | 74 MRONJ 68 ORN | DB (10) DB + BP (10) BP (41) | CT (14) Minimal debridement (1) ST (1): resection with or without reconstruction. | N.R. | In MRONJ group: higher median age, higher nr. of females, use of steroids, history of pain, periosteal reaction, and tooth extraction. Minimal debridement was significantly performed in the MBROJ group. Surgical resection was performed in the ORN group. |
| Hallmer F. et al. (2018) [40] | P | I (10) II (36) III (9) | 55 (55) Neoplasms (24) Osteoporosis (31) | 63.6 (M)- 73.1 (F) | DB (12) DB + BP (11) BP (32) | ST was sequestrectomy in patients without progressive bone disease or block resection patients with progressive bone disease with ongoing bone destruction. | 15.8 months | Prevalence of MRONJ is 3.64% in those on high-dose DB. Periodontal disease preceded development of MRONJ in 41 patients. Remission or healing in 80% of patients treated with sequestrectomy and in 92.5% of patients treated with block resection. |

**Table 2.** *Cont.*

| Authors (Year) | Study Design | MRONJ-Stage (n. Patients) | N. Total Patients (n. Lesions) and Type of Primary Disease (n. Patients) | Average Age (Years) | Types of Antiresorptive Drugs (n. Patients) and Duration of Therapy (Months, Years) | Description of Treatment (n. Patients) | Follow-Up (Months) | Outcomes |
|---|---|---|---|---|---|---|---|---|
| Ristow O et al. (2021) [41] | R | I: 86 | 55 (86) Neoplasm (46) Osteoporosis (9) | 68.2 ± 8.5 (F) 72.5 ± 10.4 (M) | BP (24) DB (25) DV + BP (5) for 49.3 ± 41.5 months | Surgical treatment: VELscope system Vx. Bone resection performed until the bright fluorescence of healthy bone was observed. Visible bone group (46 lesions): bone visibly exposed to the oral cavity. Probable bone group (40 lesions): could be probed to bone through the sinus tract. | ≤8 week | Intraoperatively, the necrotic lesions were significantly larger vs. preoperative mucosal lesions in both groups. There is a significant but very weak relationship between the extent of the mucosal lesions and the necrotic bone area. |
| Otto Sven (2016) [42] | P | I (4) II (42) III (8) | 54 Oncologic (45) Osteoporotic (9) | 71.4 ± 9.2 | DB (3) DB + BP (47) BP (4) for 46.3 months | Fluorescence-guided surgery: complete removal of necrotic bone, monitored using the visually enhanced lesion scope (Velscope), followed by smoothening sharp bony edges and meticulous wound closure. | 12.9 months | Intact mucosa in absence of exposed bone, pain, or signs of infection in 47/54 patients (87%) and in 56/65 lesions (86.2%) after first surgery. In 4 patients with 6 lesions, a second fluorescence-guided surgery was necessary. In total, 51 of 54 patients (94.4%) and 62 of 65 lesions (95.4%) showed complete mucosal healing. |
| Kojima Y. (2022) [43] | R | I (9) II (30) III (14) | 53 Neoplasms (33) Osteoporosis (22) | 74.9 ± 11.9 | DB (16) DB + BP (5) BP (32) for 47.0 ± 33.9 months | CT: oral hygiene guidance, antibacterial mouthwash, local lavage, administration of oral antibiotics, removal of mobile segments of bony sequestrum, and extraction of symptomatic teeth. | 729 ± 494 days | Clinical symptoms of 15 (28.3%) disappeared or improved while worsening was observed in 6 (11.3%). Enlargement of the osteolytic lesion occurred in 17 (32.1%) patients. CT is successful in 12 (22.6%) patients and unsuccessful in 41 patients (77.4%). Patients with stage III MRONJ have worse outcomes. The MRONJ stage and primary disease are not associated with the enlargement of osteolysis on the radiological images. A periosteal reaction on radiological examination is correlated with poor comprehensive treatment outcomes. |

**Table 2.** *Cont.*

| Authors (Year) | Study Design | MRONJ-Stage (n. Patients) | N. Total Patients (n. Lesions) and Type of Primary Disease (n. Patients) | Average Age (Years) | Types of Antiresorptive Drugs (n. Patients) and Duration of Therapy (Months, Years) | Description of Treatment (n. Patients) | Follow-Up (Months) | Outcomes |
|---|---|---|---|---|---|---|---|---|
| Eguchi (2017) [44] | R | II (52) | 52 Neoplasm (35) Osteoporosis (17) | CT: 74.8 ± 10.3 ST: 72.3 ± 11.3 | DB +/− BP (38) BP (14) | CT (24): debridement, systemic antibiotics, analgesics, and incisional drainages. ST (28): necrotic bone resection. | 6 months | CT: success in 8/24 pts (33.3%) ST: success in 25/28 pts (89.3%) |
| Blatt (2022) [45] | P | I (41) II (10) III (1) | 52 Neoplasm (52) | 71.5 ± 8.6 | N.R. | Arm A: resection of necrotic bone and fistulas; sufficient, vascularized, and mechanically stable wound coverage. Arm B: surgical treatment + PRF membrane on the decorticated bone before covering. | 6 weeks | Dehiscence and mucosal integrity after surgery in 16 cases (30.76%). No significant differences in VAS score, PWAT score, and IPR wound healing score. |
| Ahrenbog (2020) [46] | P | III | 44 Neoplasms (37) Osteporosis (7) | 68.1 | DB (4) DB + BP (8) BP (32) for 56.4 months | ST: removal of the resected bone, sharp bone edges, and the surface were smoothed till visible bleeding was reached. Afterward, wound closure was performed by using different techniques (mylohyoid muscle flap and buccal fat flap). | 13.2 months | Relapses in 12 cases and mucosal integrity in 38. Cases treated with the muscle or fat flap showed better results regarding the recurrence rate and soft tissue healing. Pain level was reduced significantly. Partial hypoesthesia of the lip arose in 18 cases. |
| Mamilos (2021) [47] | P | III (44) | 44 Neoplasm (22) Osteoporosis (22) | 68.1 | DB (4) DB + BP (8) BP (32) | ST: necrotic bone area resection or continuity resection of the mandibular. Analysis of CRP and leukocytes at baseline. | 13.2 months | The stage of chronic inflammation is correlated with the amount of vital bone and the success of surgery. If acute inflammation is dominant, chronic inflammation areas are found less, while necrotic areas are observed more. The risk of relapses, wound healing disorders, and the level of CRP are elevated if acute inflammation is severe or moderate. |
| Moll (2021) [48] | P | III (43) | 43 Neoplasm (36) Osteoporosis (7) | 68 | DB (13) DB + BP (9) BP (39) for ≥63/<63 months | Surgical treatment: necrotic bone area resection, perioperative, and antimicrobial mouth rinse with chlorhexidine. | 21.86 weeks | Significant improvement of EORTC QoL-H&N35 and the Oral Health Impact Factor-G14 (OHIP-G14) questionnaire |

Table 2. *Cont.*

| Authors (Year) | Study Design | MRONJ-Stage (n. Patients) | N. Total Patients (n. Lesions) and Type of Primary Disease (n. Patients) | Average Age (Years) | Types of Antiresorptive Drugs (n. Patients) and Duration of Therapy (Months, Years) | Description of Treatment (n. Patients) | Follow-Up (Months) | Outcomes |
|---|---|---|---|---|---|---|---|---|
| Soutome (2020) [49] | R | II (18) III (20) | 38 Neoplasms (17) Osteoporosis (21) | 74.3 | DB (14) BP (24) for 4 years | 38 MRONJ patients with PR Type 1: new bone is formed parallel to the mandible, and no gap was evident between the mandible and new bone; type 2: new bone is formed parallel to the mandible, and a gap was evident between them; type 3: an irregular shape. Segmental mandibulectomy (9) Marginal mandibulectomy (29) | ≥3 months | Inflammatory tissue in the area visualized as a gap on CT at histological examinations. Presence of bacteria in the type 2 or type 3 PR at bacteriological examination. Complete cure in 21 of 38 (55.3%) patients, lower than the cure rate of 73.4% in 143 patients without PR. Cure rate lower in cases with type 3 PR or with persistent osteolysis. |
| Fleisher (2016) [50] | R | II (31) III (3) | 31 (33) Neoplasms (31) | 64 | DB (10) DB + BP (3) BP (18) for 47 months | ST: marginal resection with a saw or bur and osteotome to resect sequestra, necrotic bone, reactive bone, and clinically uninvolved bone that was identified by FDG uptake. Low-risk group (type A): 22 patients with activity limited to the alveolus, torus, and/or basal bone superior to the mandibular canal. High-risk group (type B): 11 patients with type A FDG activity with extension inferior to the mandibular canal. | 6.6 months | Treatment of type A MRONJ lesions was more successful than treatment of type B MRONJ lesions. 7 of the type B failures were successfully retreated by segmental resection and reconstruction. |
| Klingelhöffer et al. (2016) [51] | P | I (34) II (36) III (6) | 29 Neoplasms (33) Osteoporosis (4) Rheumatoid arthritis (3) | 70.9 | 20/20 | Preoperatory antibiotics and ST | 55 weeks | Long-term maintenance of the mucosal closure in 27.6% of patients. Stage II patients decreased to stage I in 81% after surgery, and stage III patients improved in 83% of cases. Stage I patients profited only in 38% by surgical intervention. MRONJ recurrence after surgery is associated with extended preoperative MRONJ duration. MRONJ of the upper jaw seems prognostically more favorable. |

**Table 2.** *Cont.*

| Authors (Year) | Study Design | MRONJ-Stage (n. Patients) | N. Total Patients (n. Lesions) and Type of Primary Disease (n. Patients) | Average Age (Years) | Types of Antiresorptive Drugs (n. Patients) and Duration of Therapy (Months, Years) | Description of Treatment (n. Patients) | Follow-Up (Months) | Outcomes |
|---|---|---|---|---|---|---|---|---|
| Hauer (2020) [52] | R | I (3) II (24) III (5) | 26 Osteoporosis (26) | 73.4 | DB (8) DB + BP (4) BP (16) for 73.1 months | ST: radical removal of necrotic bone was performed by resection with the borders in nonvital bone, followed by removal of residual osteonecrosis by rotary burr into the viable bleeding bone margins and under VELscope control. In lesions with sequestration, the sequestrectomy is performed, and the remaining necrotic bone is then radically removed by rotary burr. | 20.5 months | Complete healing was observed in all patients, in 9% of cases by secondary intention, in the mean period of 6 weeks. |
| Hoefert et al. (2017) [53] | R | I (1) II (10) III (6) | 17 Neoplasms (16) Osteoporosi (1) | 68.5 ± 12.0 | DB 120 mg (17) DB 60 mg (2) for 19.7 ± 10.5 months | Major ST (5): sequestrectomy, bone smoothing, tension-free tissue coverage, and drainage. Minor ST (1): palatal sequestrectomy and soft tissue closure. CT: (10): surface debridement, local rinses, and intermittent antibiotics. | 348 ± 329 days | Pain at the first visit in 47% of patients, of which 24% had pressure-like pain. The majority of MRONJ are at sites of dental prostheses-induced pressure sores (41%) or dental extractions (35%). Complete healing is significant in patients treated with major ST (80%) vs. CT (20%). DB is discontinued in 60% of nonoperative patients and major ST patients with no effect on healing. Histologic findings exhibit fewer osteocyte lacunae, and micro-CT reveals trabecular thickening. |
| Pichardo (2016) [54] | R | II (2) III (9) | 11 Neoplasms (7) OsteoporosiS (4) | 72.6 | DB: 6 DB + BP: 5 For 17 months | ST: debridement with cauterization of the bone + antimicrobial treatment. | 16.4 | Healing in 9 patients |
| Beaudouin (2021) [55] | R | I (6) | 6 Neoplasm (6) | 66.5 | DB (6) | Surgical treatment: necrotic bone area reSection 5–9 months after Dmab withdrawal. Conservative treatment: oral hygiene, antibacterial mouthwashes, systemic antibiotic therapy, and not wearing overlying dentures. | 23.5 months | Dmab was stopped in MRONJ patients, with favorable outcomes for 3 cases and stabilization in 4 cases. |

## 3. Results

The electronic database search identified a total of 2657. After duplicate removal, 1776 studies underwent title and abstract screening. In total, 1704 papers were not selected after the abstract screening, and 68 articles were chosen for the eligibility assessment. Subsequently, 36 papers were eliminated after the full-text evaluation: in total, 15 were off-topic, 7 had the wrong setting, and there were 15 with no outcome of interest. Finally, 31 articles were picked for the systematic review. The selection process is summarized in Figure 7.

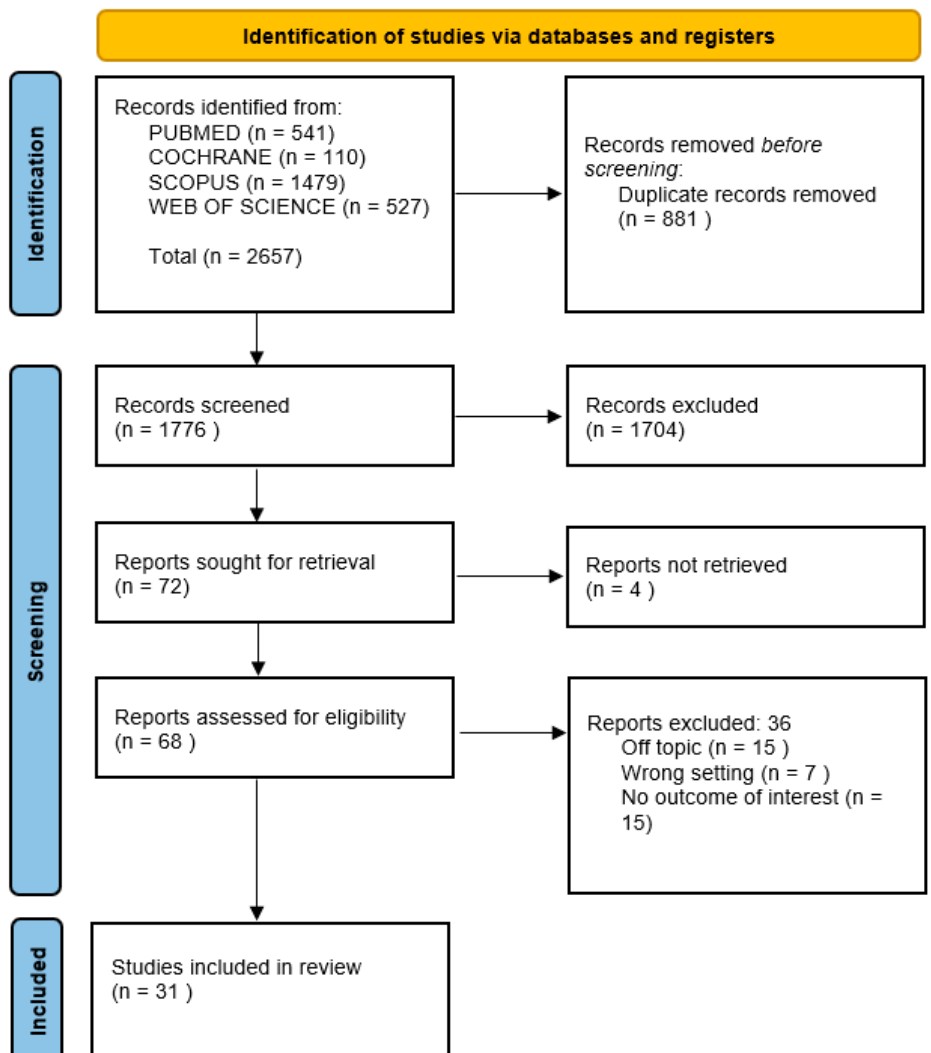

**Figure 7.** Literature search Preferred Reporting Items for Systematic Reviews and Meta-Analyses (PRISMA) flow diagram.

## 4. Discussion

### 4.1. Medical Treatment

Two types of treatment of MRONJ exist in the literature: "nonsurgical" and surgical. The first, also called "conservative," is based on the almost exclusive use of drugs to control infection and pain in order to stabilize the clinical setting, slowing disease progression; this remains the recommended treatment option in the early stages of the disease. The second approach, surgical, is reserved for advanced forms of MRONJ or those refractory to conservative treatment. A conservative approach may lead to resolution only in the early stages of MRONJ and in an otherwise limited number of cases according to the current AAOMS classification scheme, while it would be poorly effective in more advanced

stages of the disease [14,56]. In agreement with Ruggiero et al., nonsurgical treatments, consisting of antibiotic therapy and antimicrobial mouth rinses, are considered the gold standard in the management of MRONJ, and complete healing of lesions is not considered mandatory; stable lesion condition or downstaging of MRONJ, according to the AAOMS, are considered the goals of conservative treatments [14,21,31,57].

### 4.1.1. Antiseptic Therapy

The use of chlorhexidine mouthwashes (e.g., 0.12% nonalcoholic solution and 0.2% alcoholic solution) is widely recommended for disinfection of the oral cavity in the presence of oral mucosal lesions, whether resulting from spontaneous trauma or invasive dental-surgical procedures [58]. In the case of patients with MRONJ, temporary use of chlorhexidine, limited to the flare-ups of an over-infection or in the perioperative period after more or less invasive oral cavity surgery, is preferred. In this situation, the use of 0.2% alcohol chlorhexidine is recommended. Beyond this, an antiseptic maintenance protocol with nonalcoholic chlorhexidine 0.12% (2 rinses/day, 1 week/month) is suggested in individuals with MRONJ who cannot undergo therapeutic protocols with curative intent due to comorbidities or antineoplastic therapies that cannot be deferred. The purpose is to limit the emergence of bacterial resistance and the undesirable effects of chronic chlorhexidine therapy [56,59].

The use of chlorhexidine as an oral home wash has been doubted by a 2019 U.S. study, despite belonging to the current AAOMS MRONJ treatment guidelines. In fact, washing alone does not remove biofilm from exposed bone, which remains a continuous source of irritation, inflammation, and infection. In this study, home wound care therapy is presented, in which the patient is instructed to use a chlorhexidine-dipped cotton swab and to scrub mechanically on the wound. The encouraging results, with 71% resolution and 22% improvement in disease, make this practice a useful nonsurgical adjuvant treatment [32].

### 4.1.2. Antibiotic Therapy

The use of antibiotics is justified by the fact that infection is a condition that accompanies, if not determines, the clinical manifestations of the disease, and therefore antibiotic therapy plays a decisive role in the management of related signs and symptoms. The almost complete absence of RCTs on antibiotic treatment in patients with DB ONJ makes it impossible to confidently define the efficacy of individual molecules in treating this disease. The infectious component of pain in individuals with MRONJ responds well to antibiotic treatment in the early stages of the disease but tends to lose efficacy over time [60].

The protocol found in the literature involves the use of antibiotic combinations of penicillin and metronidazole, as first choice: the duration of therapy from a minimum of 7 to a maximum of 14 days, at full dose; the route of administration of choice is oral suspension; possible alternative molecules (e.g., erythromycin, clindamycin, or ciprofloxacin) are to be used in case of penicillin/cephalosporin allergies [14,21,31,58].

In the study by Schiodt M. et al. (2018) [27], the enrolled patients had ONJ caused by stage II DB intake according to the AAOMS classification and underwent conservative treatment by medication or surgery. Most patients included in the registry (92%) were treated with drugs for ONJ: In total, 80% received antibiotics, and 59% received oral rinses. Of the patients (31%; *n* = 102) whose ONJ was managed surgically, 55 patients (17%) were treated with minimal surgery (e.g., debridement, sequestration, and curettage). The most commonly used class of antibiotics was extended-spectrum penicillin (212 cycles). The doses of antibiotic therapy varied, although they reflected standard dosing regimens. Most patients required more than one course of therapy: in total, 37% received one course, 44% 2–4 courses, and 19% ≥5 courses. The average duration of antibiotic use was 28 days. The resolution of ONJ was observed in 35% of the patients (median follow-up time of 11 months); overall, more than half of the patients in the study had a resolution or improvement of ONJ [27].

The study by Kojima Y. (2022) [43] analyzes the role and effectiveness of conservative therapy and the correlation between overt signs and symptoms and lesion progression in 53 patients with MRONJ caused by DB intake. Every patient in this research received conservative treatment, which included oral hygiene instruction, antibacterial mouthwash, local lavage, and oral antibiotic administration (amoxicillin, clindamycin, and sitafloxacin). Conservative treatment options include the removal of movable bony sequestrum segments and the extraction of symptomatic teeth from the exposed necrotic bone. The results of conservative therapy were favorable in 22.7% of the patients and unfavorable in 77.3% of the patients. While MRONJ lesions are undergoing conservative therapy and seem to progress clinically, this is not always the case. Regardless of the clinical outcomes, patients, particularly those who have periosteal reactions, require ongoing CT scan monitoring [43].

In the Kaibuchi et al. (2021) paper [29], patients were initially given conservative care, including antibiotics and mouthwash (saline or povidone iodine) (penicillin compounds, cephem compounds, or macrolide compounds). Intravenous antibiotic therapy with ampicillin/sulbactam appears to generate clinically adequate bone concentrations with no significant variations between healthy and necrotic tissue; oral therapy (e.g., 875/125 mg twice a day with amoxicillin/clavulanic acid) results in significantly lower concentrations than intravenous administration [61]. The mucosal wound must be completely closed in order to be recognized as cured. By using dental tools, the result was examined visually and felt to determine its adequacy. Only 45% of the patients had to undergo sequestrectomy, and then 55% of the patients recovered with conservative treatment [29].

### 4.1.3. Pain Relief Therapy

Pain is a frequently encountered symptom in patients with MRONJ, which determines the clinical manifestation of the transition to more advanced stages of the disease. The use of NSAIDs, opioids, ketamine, neuroleptics, or others for treating chronic neuropathic pain in individuals with drug-related ONJ should be managed entirely by analgesic therapy specialists to avoid incurring intoxication (i.e., for opioids) or reduced efficacy over time [14,21,60].

MRONJ is accompanied by soft tissue pain caused by infection and inflammation, which may have all of the characteristics of nociceptive, somatic pain that causes discomfort in the more severe and acute phase [62]. Anti-infective medications are usually effective in alleviating this pain. However, there are some characteristics of MRONJ pain, particularly when mild, that may indicate neuropathic pain based on duration and resistance to therapy [62].

### 4.1.4. Teriparatide Therapy

Teriparatide is a parathormone-derived molecule used in the treatment of severe osteoporosis. Its mechanism of action is to stimulate bone production by osteoblasts. It thus has a direct anabolic effect on bone, increasing bone mass and strength, unlike BPs, which counteract bone loss by blocking osteoclast-mediated remodeling. It is now hypothesized that the use of teriparatide in pathological conditions of the jaw bones associated with alterations in bone metabolism may produce beneficial effects both in terms of reduction or resolution of bone loss and control of clinical signs of disease [63].

The Sim et al. (2020) [64] placebo RCT study examined the efficacy and safety of a 2-month teriparatide therapy in healing established MRONJ lesions over the following 12 months in patients with cancer or osteoporosis. Thirty-four participants with established MRONJ, with a total of 47 distinct MRONJ lesions, participated in the study and were assigned to 8 weeks of subcutaneous injections of teriparatide (20 mg/day) or placebo, in addition to calcium and vitamin D supplementation and standard clinical care. Participants were observed for 12 months, with primary outcomes including the clinical and radiological resolution of MRONJ lesions. Secondary outcomes included osteoblastic responses measured biochemically and radiologically and changes in quality of life. MRONJ lesions progressively resolved in both groups during the follow-up period, with 45.4% of the

lesions resolved at 52 weeks in the teriparatide group and 33.3% in the placebo group. Teriparatide was significantly associated with a higher rate of MRONJ lesion resolution than placebo [64,65].

### 4.1.5. Pre- and Postsurgery Pharmacological Protocols

Mamilos et al. (2021) reported in their study that patients were treated with perioperative i.v. antibiotic treatment with amoxicillin and clavulanic acid from one day before surgery until the 10th postoperative day. Clindamycin was administered in patients allergic to amoxicillin. In cases of acute inflammation (C-reactive protein is raised), patients may benefit from protracted preoperative antibiotic therapy to lower the acute inflammatory tissue within the bone. Reducing tissue inflammation is relevant to minimizing the occurrence of relapse [47]. Amoxicillin-clavulanic acid and extended-spectrum penicillin were the most used antibiotic classes. Usually, more than one course of therapy was received by patients, and the average duration of antibiotic administration was 28 days [27]. Patients undergoing lower cumulative dosages of ARAs (<2 years) could carry on antiresorptive treatment during invasive dental treatment [13]. Beaudouin et al. (2021) planned MRONJ surgical management 5–9 months after DB interruption, with antibiotics therapy from the day before and for at least 21 days after the surgery. DB reintroduction was performed on a case-by-case basis, also based on wound healing [55]. Hayashida et al. and Yoshida et al. assessed that despite the primary disease, it is not essential to withdraw ARAs when conducting surgical treatment of MRONJ because drug holidays showed no effect on improving outcomes [30,66]. Temporary withdrawal of DB may be an efficacious way in the future to hinder the advancement of the disease, with a perspective of treating multiple modalities of ONJ to decrease the use of invasive surgical treatments [67].

### 4.2. Laser Therapy

The application of low-intensity laser (low level laser therapy—LLLT) has been successfully reported as an adjunctive treatment in the medical or surgical management of MRONJ. The biostimulating effect of numerous wavelengths enhances reparative processes, increases the inorganic bone matrix and the mitotic index of osteoblasts, and stimulates the growth of blood and lymphatic vessels. It has also been reported that LLLT improves bone healing in traumatized sites and increases mineralization during regenerative processes after implant placement by stimulating osteoblast activity and differentiation. Laser biostimulation, which can be practiced with different wavelengths, could also be an adjunctive therapy in treating "early" forms of MRONJ, being a safe, minimally invasive, and well-tolerated technique [14].

In the study by Favia et al. (2018) [31], a group of 24 patients was selected for whom surgery was not considered completely safe or could not discontinue cancer-related therapies and, therefore, required only nonsurgical management of MRONJ. The treatment protocol included the use of an antiseptic mouth rinse (chlorhexidine), periodic dental checks, systemic administration of antibiotics (ceftriaxone 1 g/i.m. daily and metronidazole 500 mg/per os twice daily for 7 days once a month), and LLLT therapy, which consists of irradiation of necrotic bone with a diode laser employed with a 320 μm fiber, a wavelength of 800 ± 10 nm, at a power of 0.5–1 W, and removal of bone sequestrations from the surface of exposed bone. The results never showed complete healing of the lesions: in total, 87.5% of the lesions remained stable, one lesion rose from stage II to III, and only two lesions improved, with a descent from stage III to II and stage II to I, respectively [31,68].

### 4.3. Minimally Invasive Surgery: Debridement

Areas of necrotic bone are a constant source of soft tissue irritation that can progress the stage of ONJ disease and, therefore, must be removed.

Minimally invasive surgical techniques are the gold standard of treatment for MRONJ stages I and II, whereas later stages are best treated with more involved surgical techniques [27]. Debridement and sequestrectomy are two types of minimally invasive surgery. Debride-

ment, or bone curettage, is the surgical removal of necrotic bone tissue until a bleeding bone surface is found and is performed when a viable bone is attached to the necrotic bone. Typically, the defect is completely closed by mobilizing a mucoperiosteal flap, and the operation can be performed under local or general anesthesia.

Sequestectomy is the surgical procedure of removing necrotic bone sequestration, a portion gradually separated from the adjacent healthy bone. Often the sequestration undergoes spontaneous exfoliation; in some cases, it requires surgery under locoregional anesthesia or general anesthesia, depending on the extent of the process, the clinical condition, and the patient's compliance [29]. A Belgian study performed on a sample of patients, of whom 43 had taken DB, investigated the efficacy of minimally invasive surgical techniques in MRONJ. Its results, with 73% of the lesions in complete healing or downstaging after 24 months of follow-up, encourage the use of minimally invasive surgical techniques early to avoid the evolution of the disease into advanced stages, and, in most cases, a regression can be expected [36]. According to Akashi's study, surgical management of MRONJ appears to try to prevent local infection and alleviate pain as a substitute for frequently giving patients analgesics, especially elderly ones [39]. The auxiliary use of PRP in minimally invasive techniques was observed in the study of Coropciuc and Sánchez-Gallego and was held responsible for a significant improvement [36,38]. Some observational studies agree that surgical outcomes of ONJ vary based on the type of baseline antiresorptive therapy: patients who received DB alone had higher rates of resolution and lower rates of progression than those who received zoledronic acid alone [27,33]. Moreover, Wei's study concludes that drug type is a prognostic indicator for MRONJ and that DB could outperform zoledronate to treat bone metastases in prostate cancer patients [33,69].

*4.4. Surgical Resection*

The surgical approach to drug-related ONJ is one of the most analyzed topics in the literature, and the type of treatment is determined according to the clinical staging of the disease. In Favia's publication, a surgical protocol with piezosurgery based on size staging is presented: in stage II, a small open access of the bone margins is suggested; in stage III, piezosurgery of the bone margins is combined with the application of hyaluronic acid and amino acids [21]. According to the study of Osaka, extensive surgery, consisting of the removal of the necrotic and surrounding bone, marginal mandibulectomy, or partial maxillectomy, is recommended for MRONJ stage II and III [34]. Watanabe declared that extensive surgery is highly effective against MRONJ regardless of the stage of the disease [28].

According to the most recent literature, the primary intent of surgical therapy in drug-related ONJ is not to be considered palliative but curative; it is achieved through the complete removal of the tissue macroscopically involved by the disease, leaving only healthy tissue to allow stable healing over time. Moll et al. evaluated how much the quality of life changed after surgery, considering the stage of the disease, the impact of the surgery, and recurrence: surgery significantly improves the quality of life of stage III patients, including the quality of life of those who suffer a recurrence, resulting in MRONJ stage I [48].

Surgery will be less invasive and have a greater margin of success if the disease to be treated is limited in extent [40,53]; therefore, reserving surgery for the more advanced stages of the disease, as advocated in past years, is no longer a desirable option, especially considered the poor results obtained with conservative treatments, even in the early stages of ONJ [37]. El-Rabbany states the superiority of surgical treatment over conservative treatment in MRONJ patients [37]. In a prospective cohort study on 11 patients, Hallmer observed healing or remission in 80% of the patients undergoing sequestrectomy and 92.5% of the patients undergoing resective en bloc surgery [40].

The tissue directly affected by drug-related ONJ from its early stages is bone; the complete removal of the involved bone tissue should lead to the resolution of the clinical problem, and the removal of the involved soft tissue should be unnecessary. The presence of histologically healthy bone tissue at the bone resection margin allows complete and stable



healing [54]. According to Suutome, surgical treatment is more effective than conservative treatment; however, healing is more successful in cases with no periosteal reaction, as measured using CT, histological analysis, and bacteriological examinations [49]. The Mamilos study demonstrates the existence of a correlation between the failure of resective surgery and the severity of an acute infection in patients with MRONJ stage III. The authors suggest prolonged antibiotic prophylaxis with amoxicillin and clavulanic acid before and after surgery [47].

For the complete removal of the pathological bone tissue, it is necessary to identify the healthy tissue surrounding the lesion with a good margin of safety [50].

In current clinical practice, there are two orientations: the first is based exclusively on the intraoperative determination of the resection margins, while the second aims to identify the true extent of the pathological tissue prior to surgery using radiological methods. Evaluating bone bleeding is still the most widely used means of intraoperative identification of surgical margins in MRONJ. In order to increase the predictability of intraoperative assessment of necrotic and still vital bone, a fluorescence lamp has been proposed in the recent past to facilitate its recognition [50].

In the literature, there is still a considerable disparity in the evaluation of the effectiveness of treatments for drug-related ONJ, which originates from a variable definition of the evaluation criteria [21,70]. The most widely used definition of healing, for example, is based on clinical criteria, such as maintenance of the seal of the oral mucosa without any symptoms. This definition is largely incomplete, as it completely misses the assessment of the bone, which is where the disease develops and may recur over time, regardless of the persistence of the overlying mucosal seal [21,34,53].

The term 'resective surgery' refers to the en bloc removal of pathological bone down to healthy tissue [54].

Unlike oncological surgery, bone resection margins in drug-related ONJ, as in all forms of osteonecrosis and osteomyelitis of the jaws, are not codified [50]. Preoperative TC and RM evaluation of the resection margins allows for identifying the adjacent normal bone tissue with good accuracy, which, if nonpathological on histological examination, ensures complete and stable healing over time [40]. There are two forms of resective surgery: marginal and segmental. Marginal resective surgery consists of the en bloc removal of pathological tissue without interruption of the anatomical continuity of the skeletal segment concerned. This surgery can be performed under locoregional anesthesia or general anesthesia, depending on the extent of the process, the clinical conditions, and the patient's compliance [47].

Segmental resective surgery refers to the en bloc (full-thickness) removal of a skeletal segment with interruption of its anatomical continuity [54]. This surgery is performed under general anesthesia. It must always include an additional osteoplasty of the resection margins to eliminate possible residual asperities and mucosal closure by the first intention of the defect, by mobilizing a mucoperiosteal flap if no other form of reconstruction is indicated [26].

In the treatment of advanced stages of ONJ, the high full-thickness mucoperiosteal flap was prepared after a horizontal incision to the periosteum. It was fixed with resorbable suture material within the periosteum for the closure of the bone defect. However, in cases of oral soft-tissue breakdown, the mylohyoid muscle flap (MMF) in the lower jaw, and the buccal fat flap (BFF) in the upper jaw, were reliable methods to have better coverage of the bone defects and healing of tissues [38,71]. Ahrenbog et al. evaluated the use of MMF and BFF as additional tissue closure in stage III cases of AAOMS classification, compared to mucoperiosteal flap closure alone. Mylohyoid muscle was carefully detached from the mouth base mucosa to have complete mobilization, and MMF was performed for tissue closure after the resection of osteonecrosis of the lower jaw. BFF was separated from the lateral pterygoid muscle and was used in the lateral upper jaw to cover the defect in cases of oroantral communication. The flap was fixed at the periosteum with absorbable suture material for holding it in position. These flaps ensured better results than the

mucoperiosteal flap alone and were a good alternative to the usual procedure of soft tissue closure in stage III cases of MRONJ [46]. Similar outcomes were found in the study of Ristow et al., which observed the advantages of MMF and BFF in stage I, stage II, and stage III cases of AAOMS classification. Double-layer closure techniques after surgery in MRONJ increased the chances of tissue healing due to increased mechanical stability and better vascularization of the flaps covering the bone defect. MMF and BFF were only used if the anatomical sites were suitable and, with a low rate of complications, turned out to be very convenient for tension-free wound closure through MMF, BFF, and mucoperiosteal flap to guarantee wound closure [35].

The standard resective surgery for the mandible is mandibulectomy. It always causes the loss of symmetry of the lower third of the face and occlusion. Reconstruction after surgery can be achieved with titanium reconstruction plates to replace the removed bone with anatomical mandibular replicas or, alternatively, with vascularized bone flaps. Hospitalization time and the recovery of normal function are faster when the mandible is reconstructed with titanium plates or mandibular prostheses than with vascularized bone flaps [54].

The segmental resective surgery for the upper jaw is the maxillectomy, which is usually classified according to the vertical and horizontal extensions of the defect that is created to remove the pathological tissue [54]. A distinction is made between partial and total maxillectomies, and the first is characterized by the removal of only the dentoalveolar process, with or without preservation of the palate; the second is denoted by the en bloc removal of all bone sides, including the orbital floor [26,54].

*4.5. Fluorescence-Guided Bone Surgery*

Successful therapy should aim to eliminate bone exposure and restore mucosal integrity. MRONJ should be removed even if only small bone areas are affected because the infected necrotic and exposed bone will not be revitalized and resurrected. As a result, the goal of surgical therapy should be to remove the necrotic bone completely [42]. However, even among those who support surgical therapy, there is disagreement about which surgical technique is more effective. Indeed, one of the challenges and limitations of MRONJ therapy is that the margins of the osteonecrosis cannot be precisely determined, making clear demarcation of the necrotic bone difficult, if not impossible. Fluorescence-guided bone surgery has yielded promising results in the surgical management of MRONJ. This technique may help to define the transitions between necrotic and non-necrotic bone during the surgical procedure by providing a controllable therapeutic approach. Because this surgical approach is simple and reproducible, it may help to objectify surgical MRONJ therapy, implying an improvement in treatment [42].

It is difficult to differentiate between viable and necrotic bone, and intraoperative visualization of fluorescent patterns of viable and nonviable bone (fluorescence-guided surgery) may improve surgical outcomes. Viable bone has bright greenish fluorescence during this examination, whereas necrotic bone has no or only pale fluorescence [52].

Regarding this procedure, Otto S. et al. conducted a study in which patients were randomly assigned to undergo fluorescence-guided surgery for medication-related osteonecrosis of the jaw [42]. The first surgical intervention, using fluorescence-guided bone surgery, resulted in complete mucosal healing in 87% of the evaluated patients, and 86.2% of the lesions had no bone exposure and no complaints at the time of the last follow-up. In total, 3.7% of the patients had no complaints, no bone exposure, and complete mucosal coverage of the bone. When the first and second surgeries were combined, 94.4% of the patients and 95.4% of the lesions showed complete mucosal healing and no bone exposure. They concluded that fluorescence-guided bone resection is a safe surgical treatment option for patients with medication-related osteonecrosis of the jaw [42].

Similarly, Hauer et al. performed a monocentric reevaluation of MRONJ surgical therapy. The fluorescence of tetracycline antibiotic bound in bone tissue was used to distinguish between viable and necrotic bone during intraoperative exposure to VEL scope

(visually enhanced lesion scope) light of 400–460 nm wavelength. In all patients (100%), complete healing (complete mucosal closure) was achieved by secondary intention in 9.4% of the cases (*n* = 3) in a mean period of 6 weeks [52]. There was no need for a second surgery to achieve complete healing. Therefore, they stated that the surgical therapy protocol, which includes fluorescence-guided bone surgery, is effective in managing all MRONJ stages. The surgery should preserve as much viable bone and soft tissue as possible, with no preventive extension with safety margins required [52,72].

### 4.6. Platelet-Rich Plasma (PRP), Platelet-Rich Fibrin (PRF), Concentrated Growth Factor (CGF), and Piezosurgery (PZ)

The literature indicates high recurrence/dehiscence rates in patients with MRONJ after surgical resection, with increased hospitalization and reoperation [73]. As a result, many techniques for optimizing therapy have been undertaken, such as flap design modification or intraoperative imaging with fluorescence-guided bone surgery, but have yet to be transferred into clinics [35,74]. Another strategy that might be used is the use of autologous preparations made from the patient's blood, platelet-concentrate products, such as platelet-rich plasma (PRP), platelet-rich fibrin (PRF), and concentrated growth factor (CGF), that release large amounts of growth factors playing an important function in bone biology by speeding up and improving bone repair or regeneration [75–79]. Autologous platelet concentrates (APC) are employed in various fields of dentistry since their role in bone and soft tissue regeneration has been demonstrated [80–82]. APC induces target cells to synthesize growth factors (GF) such as TGF-β1, PDGF-BB, VEGF-A, and IGF-I which influence important processes involved in tissue healing, chemotaxis, cell proliferation, differentiation, and extracellular matrix production [83,84]. They promote healing by bringing in leukocytes, increasing collagen production, generating anti-inflammatory agents, and beginning internal vascular development [85]. The amount and rate of GF release have been demonstrated to differ between CGF, PRF, and PRP: PRP enables faster delivery of GFs to the target site, even if using PRF or CGF leads to a considerable increase in GFs, compared to PRP [86–88]. PRP, PRF, and CGF have been utilized therapeutically for various purposes, including the treatment of MRONJ, and have shown extremely promising outcomes in several trials [89,90]. According to researchers, APCs might help cure osteonecrosis by increasing patients' quality of life and decreasing pain and postoperative infections [91–93]. Including PRP in surgical therapy for MRONJ patients tends to improve the recurrence rate, with good healing in 85–90% of instances [94,95]. The use of PRP has been proposed in several studies to treat osteonecrosis caused by BPs. In a study by Curi et al., patients with BRONJ were treated with surgical necrotic bone resection and PRP discovering that full wound healing was achieved in most patients and that the BRONJ treatment duration was reduced [96]. Bocanegra-Perez et al. used PRP in the surgical treatment of BRONJ, allowing quicker mucosal healing, less analgesic use, and better clearance of oral lesions [97]. Moreover, the study of Sarkarat et al. showed positive results in terms of preservation or regeneration of bone with the use of PRP in BRONJ treatment [75]. PRP has been proposed as a first-generation platelet concentrate, but anticoagulants have been shown to interfere with platelet-mediated angiogenic and regenerative responses [98]. Choukroun developed a second-generation platelet concentrate, termed platelet-rich fibrin (PRF), that required no manipulation after blood collection and centrifugation as a substitute for PRP [99,100]. The fibrin clot's slow and continuous release of various proangiogenic growth factors and cytokines promotes the proliferation and differentiation of osteoblasts, endothelial cells, chondrocytes, and fibroblasts, improving soft tissue and bone regeneration [101]. So far, literature regarding the use of PRF in the surgical treatment of MRONJ is scarce. Blatt et al. compared the surgical treatment according to current guidelines to the addition of PRF in patients with stage I–III MRONJ, finding no statistically significant differences between the two groups in terms of wound healing, disease downstaging, pain reduction, and quality of life [45]. A different result has been reported by Nørholt et al. that used PRF for the surgical treatment of osteonecrosis of the jaw, showing its use in grade 2 ONJ may

be a factor In favorable outcomes [102]. In the study by Albertos et al., seventy patients with MRONJ were treated surgically with local debridement and PRP placement, finding a recurrence rate of 18.6% of the patients [38]. In a randomized control trial of patients with stage II and III MRONJ, Giudice et al. compared a control group to which surgical removal of necrotic bone was performed and an experimental group in which surgical removal with the addition of PRF was performed. After 1 month, faster wound healing and a lower risk of infection at the surgical site were found in the PRF group. The VAS score was also considerably lower in the PRF group, and a significant improvement in the quality of life of patients using high-dose drugs was obtained with the use of PRF after surgery compared with the control group [91]. Inchingolo et al. treated MBRONJ patients with surgical curettage and PRF and evaluated the outcomes utilizing clinical and histological techniques. They concluded that PRF could function as an efficient barrier membrane between the alveolar bone and the oral cavity and that it may provide a quick, simple, and successful alternative strategy for closing bone exposure in MRONJ patients [103]. The study of Coropciuc et al. assessed the efficacy of MRONJ conservative medicinal therapy and minimally invasive surgical treatment. Sequestrectomy, soft tissue debridement, and the administration of leucocyte- and platelet-rich fibrin were all part of the minimally invasive surgical therapy. The roles of leucocytes in platelet concentrates include anti-infective activity, immunological control, and the ability to create huge amounts of vascular endothelial growth factor [36]. CGF is the most recent generation of platelet concentrate products containing more growth factors than PRF. Yüce et al. stated that local injection of CGF appears to be a useful strategy for the surgical treatment of MRONJ in osteoporosis patients by enhancing tissue regeneration [104]. CGF has more adhesive and tensile strength, as well as greater viscosity, than PRF and PRP, and the varying centrifugation speeds allow CGF to have a larger, denser, and richer fibrin matrix. Given these advantages, CGF might be useful in treating MRONJ patients by accelerating bone and soft tissue repair [105].

Some studies have combined the use of PZ with PRP to promote soft tissue regeneration, reduce the risk of infection, increase patient comfort, and eliminate necrotic bone [106,107].

PZ is a novel surgical tool that employs ultrasonic vibrations for cutting and is used to execute several bone surgical operations in the oral and maxillofacial fields with excellent outcomes in terms of complication reduction [108].

This technique allows for a more precise, selective, and safe cut by reducing the risk of hemorrhage and damage to important anatomical structures such as nerves and membranes. It reduces stress in the alveolar bone and favors the action of repair cells in the postoperative period. Furthermore, cell survival, bone deposition, and remodeling following osteotomy appear to be improved using PZ as compared to traditional drills. Furthermore, it guarantees great visibility in the operating room, thanks partly to low bleeding, high-brightness LED lighting, and excellent irrigation [109]. In a series of 20 cases treated by Blus et al., patients with MRONJ were subjected to osseous surgery with ultrasound pharmaceutical therapy. The healing was obtained in all patients and could be confirmed over a period of 4.5 years [110]. In the study of Moll et al., after the dissection of a mucoperiosteal flap, necrotic bone was resected with a bone saw and piezo surgery. They assessed that surgery improved the quality of life of patients with stage III [48]. Furthermore, Patel et al. showed that on a total of ten patients treated just with piezoelectric debridement, without flap, eight patients were cured after six months [111].

### 4.7. Vascular Endothelial Growth Factor (VEGF) and Hyaluronic Acid (HA)

Intravenous ARA therapy determines a significant reduction in serum levels of vascular endothelial growth factor (VEGF) and TGF-β, the main reason for the onset of MRONJ [112,113].

Studies have found that local administration of VEGF stimulates a systemic increase in VEGF with reduced levels of inflammatory cytokines 1L-1α and IL-β, which are present during inflammatory processes and in bone resorption [114,115].



The proangiogenic action with the formation of new microcirculation, osteoblastic, chemotactic, and proliferative activation of VEGF, and an immunomodulatory mechanism, reduces the likelihood of osteonecrosis by promoting faster healing of the postextraction alveolar site [116,117].

HA, a glycosaminoglycan, which is biocompatible and hydrophilic, has been shown to facilitate tissue healing processes. HA combined with bone grafts of various kinds (calcic triphosphate, collagen, and autologous bone grafts) and/or osteoconductive materials enhances bone healing [118–120].

VEGFs encapsulated in an HA hydrogel solution are released over a slow time into the necrotic site of the alveolus. Associated with the reparative abilities of HA result in total healing of the necrotic bone and reduction in inflammation [121,122].

### *4.8. Ozone Terapy*

Data show that administering ozone in the gas form before and after dental treatments improves surgical and pharmacological outcomes in patients with ONJ [123,124].

Ozone, in fact, determines positive effects on bone injury because it activates endogenous antioxidative systems by influencing oxygen metabolism, acting as an antibacterial, and activating physiological reparative processes [125,126].

In patients with stage I and II MRONJ, ozone therapy has shown complete healing or significant remission of necrosis in the affected maxillary site, with spontaneous expulsion of bone sequestration and/or bone neoformation around the necrotic area, without the need for resective surgery [127].

Treatment consists of careful curettage of the necrotic site, alternating washes with saline and 10% hydrogen peroxide, and application for 8 min of ozone, in the form of an oily suspension equal to the size of the maxillary osteonecrotic lesion, after silicone impression of the arch. The number of applications performed and recommended is 10 [128].

The results demonstrate the efficacy, nontoxicity, and ease of application of ozone in oil suspension directly on osteonecrotic lesions ≤2.5 cm, making MRONJ a manageable and treatable problem [127].

### 5. Case Reports

In this part of this paper, we will show two case reports to analyze the surgical techniques discussed in this review. All cases were treated after 8 months of ARA administration suspension.

### *5.1. Case Report 1*

A 61-year-old female patient developed MRONJ after the administration of intravenous ARA. The lesion has been radiologically identified through orthopantomography (OPT) and CBCT (Figures 8 and 9).

The lesion is stage I, according to Favia's classification (2014), with bone exposure < 2 cm and the presence of spontaneous pain. The patient underwent an initial phase of systemic medical therapy with cycles of intramuscular ceftriaxone associated with orally administrated metronidazole and anti-inflammatories; topically, chlorhexidine 0.20% rinses and applications of Aminogam gel were administered.

After three cycles of drug therapy, the lesion did not heal, but the pain regressed; for this reason, it was decided to proceed with marginal resective surgical treatment. Another cycle of medical and topical therapy was administered immediately after surgery.

The surgical approach was performed using piezosurgery (Figure 10). In order to expose the lesion and improve intraoperative visibility, a full-thickness flap was performed (Figure 10A). The lesion was demarcated using the OT7 Mectron® tip, and the bone plug was removed (Figure 10B,C).

In order to promote bone regeneration, PRF and PRF autologous membranes were used to induce healing (Figure 11).

Follow-ups were performed at 7 days (Figure 12A) and 14 days (Figure 12B), during which proper mucosal healing was found as expected. Clinical and radiographic follow-ups will be performed with rx OPT at 3 months and 6 months and CT scans at 1 year.

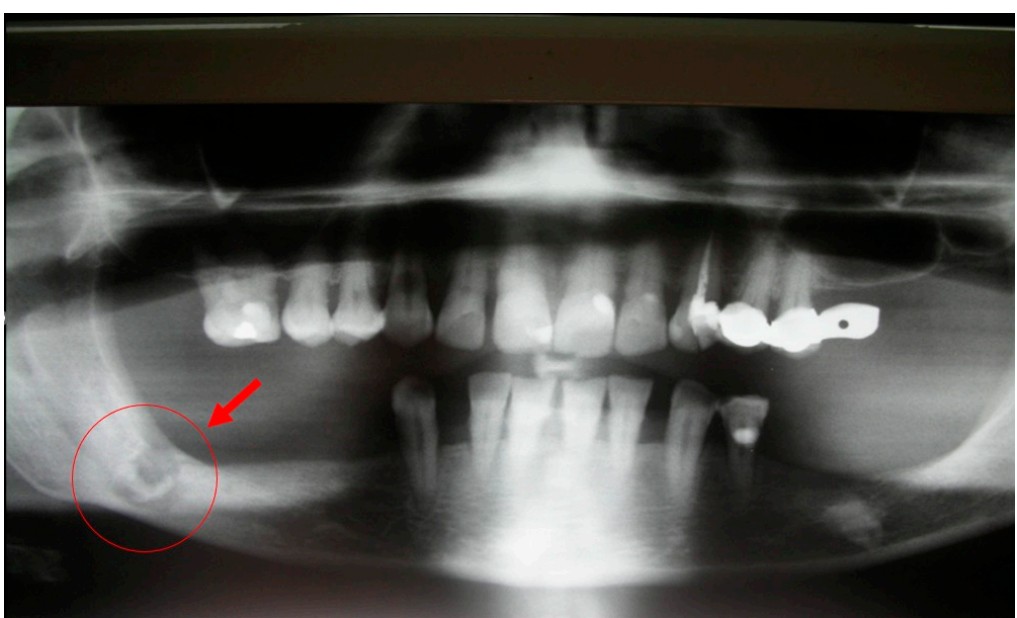

**Figure 8.** Initial lesion showed in OPT pointed by the circle.

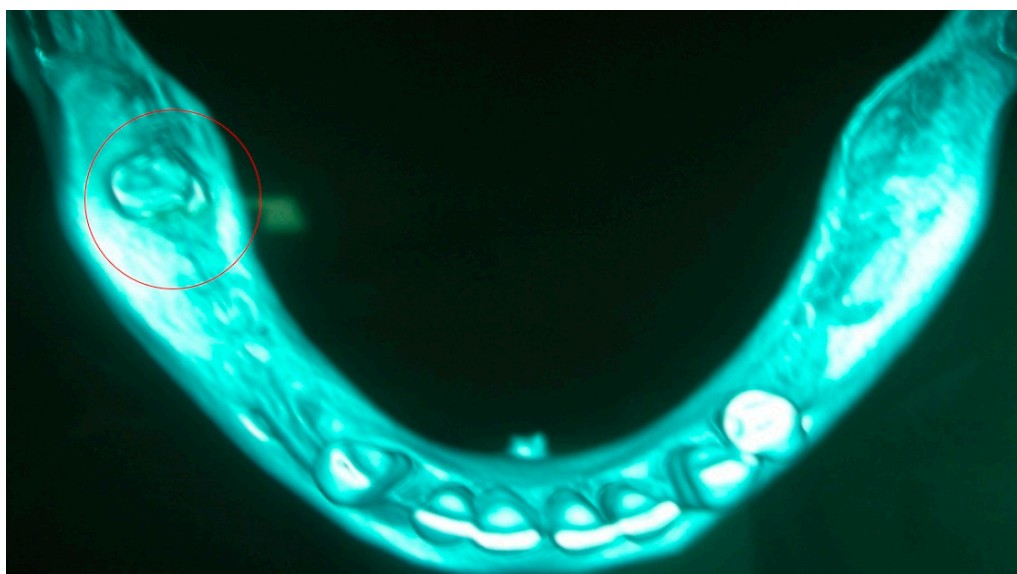

**Figure 9.** CBCT of the initial lesion pointed by the circle.

### 5.2. Case Report 2

A 65-year-old female patient developed MRONJ after the administration of intravenous DB. The lesion has been radiologically identified through OPT and is located in teeth 36–38 (Figure 13).

The lesion is stage II according to Favia's classification as it has a 3.5 cm nonpainful bony exposure. Three cycles of systemic and topical drug therapy were performed with intramuscular ceftriaxone associated with orally administrated metronidazole and chlorhexidine 0.20% rinses and Aminogam gel applications.

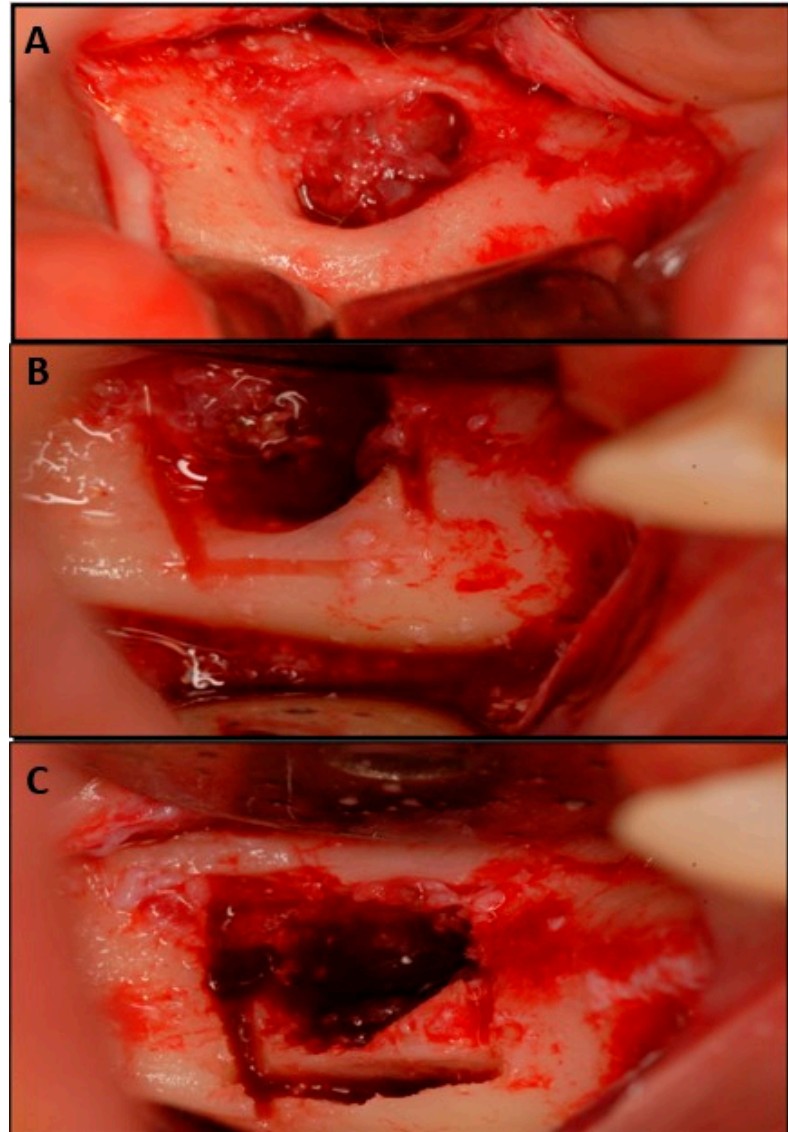

**Figure 10.** (**A**) Surgical site after incision; (**B**) Delimitation with piezo surgery; (**C**) Bone plug removed with piezo surgery only.

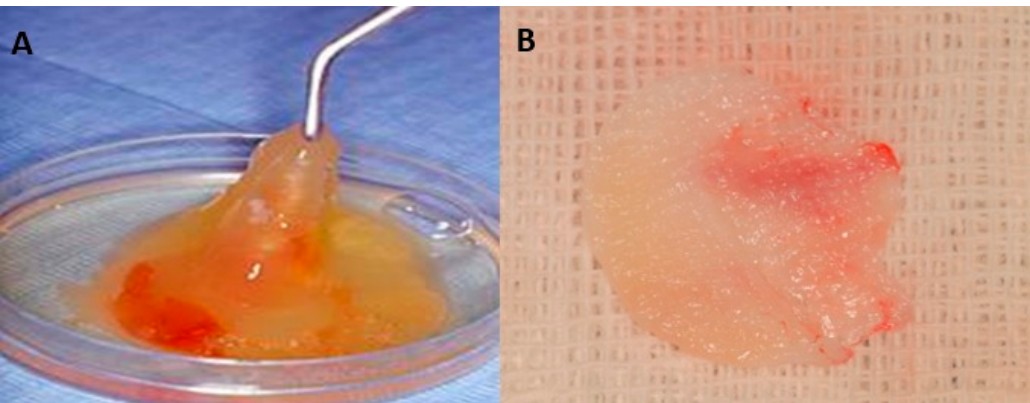

**Figure 11.** (**A**) PRF preparation; (**B**) PRF autologous membrane.

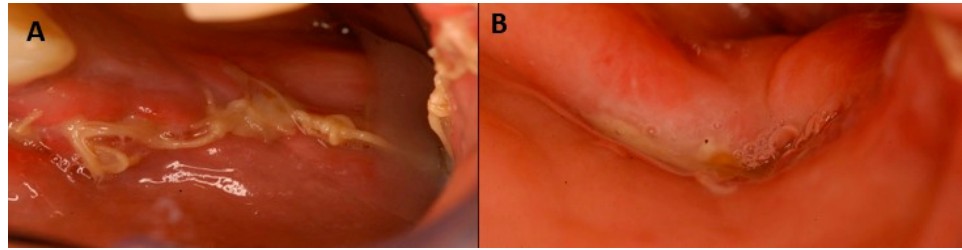

**Figure 12.** (**A**) Surgical site after 7 days; (**B**) Complete healing after 14 days since surgery.

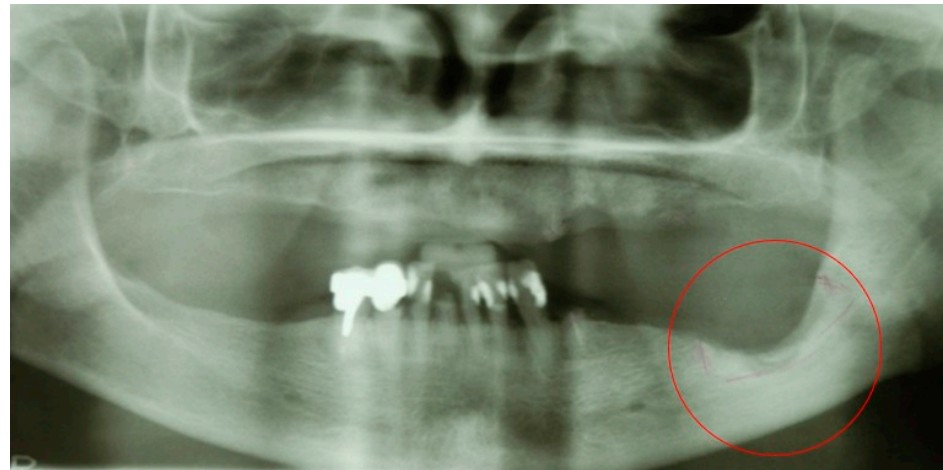

**Figure 13.** OPT of initial lesion pointed by the circle.

It was then decided to intervene surgically, using piezosurgery. In order to expose the lesion and to improve intraoperative visibility, a full-thickness flap was performed. The lesion was demarcated using an OT7 Mectron® insert, and the bone plug was removed. The edges were shaped with a PL3 Mectron® insert to facilitate healing. Bleeding bone is an important sign of tissue viability and represents for the clinician a margin for bone resection (Figure 14). Vicryl 3.0 detached stitch suture was performed, and healing was monitored at 7 (Figure 15A) and 14 days (Figure 15B); at 14 days, mucosal healing was complete.

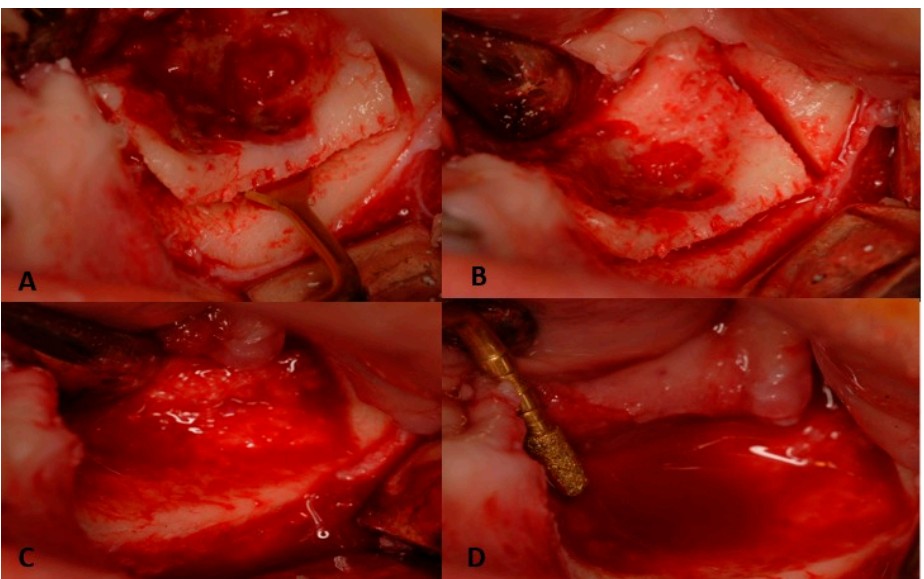

**Figure 14.** (**A**) Piezo surgery intervention with OT7 Mectron® tip; (**B**) Delimitation of necrotic area; (**C**) Removal of necrotic bone plug; (**D**) Rounding of lesion margins with PL3 Mectron® tip.

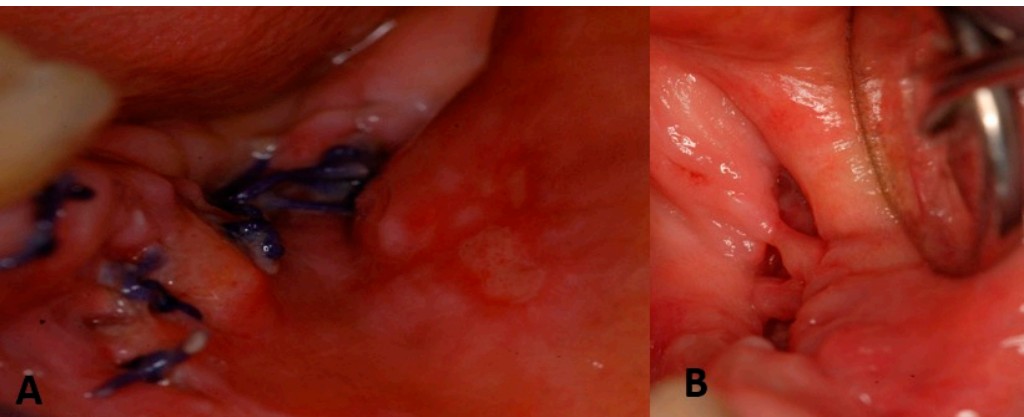

**Figure 15.** (**A**) Control of surgical site after 7 days; (**B**) Control of surgical site after 14 days (fully healed).

The patient will be monitored with clinical and radiographic follow-up with rx OPT at 3 months and 6 months and CT scans at 1 year.

In both treated cases, PRP and PRF were chosen to promote healing and decrease the risk of recurrence, as described in the literature.

## 6. Limitations

Currently, the mechanisms of MRONJ are not well understood, and the prevention and management of MRONJ are still a challenge. Therefore, it is significant to identify effective approaches to manage this well-known complication of antiresorptive drugs.

This study has a number of limitations that need to be acknowledged, such as heterogeneous patients with different systemic diseases, comorbidities (e.g., type 2 diabetes), tumors, osteoporosis, and different osteonecrosis disease stadia. The approach used in medical and surgical treatment is not standardized and is based more on clinical practice than guidelines. It was custom-made primarily for the clinicians for the single case.

## 7. Conclusions

The study of MRONJ has been of particular interest in the scientific literature for several years now. During that time, various treatment protocols have been developed, divided into surgical and nonsurgical, or more correctly, invasive and noninvasive approaches.

The studies collected in our review show that conservative treatments are effective in stabilizing the clinical picture and slowing the progression of the disease and are particularly indicated for lesions at an early stage.

Topical antiseptic therapy appears to be a valuable ally in managing over-infection or the perioperative period, but the usefulness of chlorhexidine at home is questionable. As the infectious picture is decisive for the clinical manifestations of the disease, antibiotic therapy plays a fundamental role in treating MRONJ. The most commonly used classes of antibiotics are broad-spectrum penicillins and metronidazole.

Antibiotics are also effective in the early stages of treatment for pain management, but their effect loses effectiveness over time. To control this symptom, it is advisable to consult specialists in analgesic therapies.

In recent years, the off-label use of teriparatide has been proposed for the pharmacological treatment of MRONJ. However, there are still few studies evaluating the efficacy of treating these lesions, so it will be the task of future scientific publications to investigate the potential of this drug further.

Minimally invasive debridement surgery also falls under the umbrella of conservative treatment. Surgical removal of these tissues by debridement or bone curettage represents the gold standard in stages I and II MRONJ. Follow-up of these treatments has shown signif-

icant efficacy in inducing complete healing or downstaging of the lesion, thus encouraging early recourse to prevent lesion progression.

In contrast, the efficacy of LLLT for the treatment of MRONJ lesions is questionable.

Resective surgery is still the most widely analyzed option in the literature. This approach aims not to be considered palliative but curative. The complete removal of the tissue involved in the lesion significantly improves patients' quality of life, especially in the more advanced stages of MRONJ progression.

Some authors argue that, in light of the limited effectiveness of conservative treatments, resective surgery is necessary even in the early stages of the disease. Preoperative CT or MRI evaluation of the resection margins makes it possible to identify the adjacent healthy bone tissue that, if nonpathological on histological examination, represents the margin of intervention to determine complete and stable healing over time.

In addition to the preoperative assessment, it is necessary to identify the margins of the necrotic lesions to be removed during surgery. Some authors, therefore, resort to fluorescence-guided bone surgery, which makes it possible to limit removal to necrotic tissue only, sparing as much as possible the healthy vital bone tissue at the margins of the lesion, which is essential for postoperative healing.

Although the effectiveness of surgical treatments has been amply demonstrated in the literature, the recurrence/dehiscence rate, according to some authors, is still high after lesion resection. In recent years, the scientific literature has focused on the search for techniques that could optimize therapy. Among these, the use of autologous preparations (PRP, PRF, and CGF), VEGF, HA, and ozone therapy appears promising.

Some authors also evaluated the potential of PZ, which showed considerable advantages, both from the point of view of clinical use and from a biological point of view.

**Author Contributions:** Conceptualization, A.M.I., A.P., I.F. and A.M.C.; Methodology A.P., I.F., F.V., A.N. and G.D.; Software, D.A., A.D.I., C.G.I. and E.d.R.; Validation, G.M., A.M.C., M.C., P.A. and A.M.; Formal analysis, A.P., A.D.I., A.C. and A.M.; Resources, A.M.I., E.d.R., F.V. and D.A.; Data curation, I.F., F.V., A.N., G.M. and F.I.; Writing—original draft preparation, I.F., A.M.C., M.C., P.A. and A.M.; Writing—review and editing, A.P., A.N., F.I. and G.D.; Visualization, C.G.I., A.M.I., G.M. and D.A.; Supervision, A.C., F.I., A.D.I. and G.D.; Project administration, F.I., A.D.I. and G.D. All authors have read and agreed to the published version of the manuscript.

**Funding:** This research received no external funding.

**Institutional Review Board Statement:** Not applicable.

**Informed Consent Statement:** Informed consent was obtained from the subjects involved in the study. Written informed consent has been obtained from the patient to publish this paper.

**Data Availability Statement:** The data and images used in the current study are available from the corresponding author upon reasonable request.

**Conflicts of Interest:** The authors declare no conflict of interest.

## Abbreviations

| | |
|---|---|
| AAOMS | American Association of Oral and Maxillofacial Surgeons |
| ARONJ | Antiresorptive agent-related osteonecrosis of the jaw |
| ARAs | Antiresorptive agents |
| APC | Autologous platelet concentrates |
| BRONJ | Bisphosphonate-related osteonecrosis of the jaw |
| BPs | Bisphosphonates |
| BFF | Buccal fat flap |
| CRP | C reactive protein |
| CONJ | Chemo-osteonecrosis of the jaws |
| CGF | Concentrated growth factor |
| CT | Conservative treatment |

| CTX | C-terminal telopeptide of type 1 collagen |
| DB | Denosumab |
| DRONJ | Denosumab-related osteonecrosis of the jaws |
| FDG PET-CT | Fluorodeoxyglucose positron emission tomography with computed tomography |
| HA | Hyaluronic acid |
| HBO | Hyperbaric oxygen; IPR: inflammatory, proliferative, remodeling |
| LLLT | Low level laser therapy |
| MRONJ | Medication-related osteonecrosis of the jaws |
| MMF | Mylohyoid muscle flap |
| NBPs | Non-aminobisphosphonates |
| NSAIDs | Nonsteroidal anti-inflammatory drugs |
| N.R. | Not reported |
| OHIP-G14 | Oral health impact factor-g14 |
| ONJ | Osteonecrosis of the jaws |
| OPG | Osteoprotegerin |
| ORN | Osteoradionecrosis |
| PR | Periosteal reaction |
| PZ | Piezosurgery |
| PRF | Platelet rich fibrin |
| PRP | Platelet-rich plasma |
| P | Prospective |
| RCT | Randomized clinical trial |
| RANK | Reactive activator of nuclear κb |
| R | Retrospective |
| ST | Surgical treatment |
| TKIs | Tyrosine kinase inhibitors |
| VAS | Visual analog scale |
| VEGF | Vascular endothelial growth factor |

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
