# Peer review of "MRONJ Treatment Strategies: A Systematic Review and Two Case Reports"

_applsci, doi:10.3390/app13074370_

Round 1

Reviewer 1 Report

The authors present a broad and exhaustive review on medication-related osteonecrosis of the jaw, and it is illustrated with the therapeutic approach of two cases. It is a topic that is constantly being updated and represents a challenge in the proper management of patients affected by MRONJ, so I find their work very relevant, however, the authors should correct several aspects of their manuscript, which I mention below:    

1. In Table 2, I am unable to identify an order or a weight variable that promotes the organization of the table, so it becomes unfriendly, unintuitive, unhelpful. I suggest to the authors to organize it, it could be by type of study (P or R), by number of patients (from most to least number), by best results, by stage, etc. But an order that is useful for the reader is necessary.

2. The authors mention that they found 32 articles, however, in Table 2, I only identify 31 articles, the authors could clarify why they did not incorporate that missing article or correct me.

3. Although, in Table 2, the authors mention the author and year of publication in the first column, they should also assign the citation number that corresponds to each author.

4. The image in Figure 8 appears to be out of focus or with low resolution, and some letters are underlined in red and blue. The authors should correct the details of this image.

5. After an exhaustive review and discussion, it does not seem coherent to me that the two cases reported are dealt with so tersely, especially that the approach selected for them is not justified. Although they briefly develop a justification in the limitations section, it is not enough. In the presentation of the cases, the authors should clarify which stage was assigned, under what considerations this stage was assigned, and based on their review, why they decided to use the selected approach (Section 4.6) and not another. Finally, they should acknowledge whether these were the results they expected and what their follow-up plan will be. It is important that the authors resolve these observations, because by illustrating such a broad review with these two cases, they are suggesting that this is the approach that you favor; if this is the case or not, you should justify or discuss it.

Reviewer 2 Report

 This article is very interesting. The work is well done

Reviewer 3 Report

Reviewer

 Manuscript ID: applsci-2286052
Type of manuscript: Systematic Review
Title: MRONJ treatment strategies: a systematic review and two case reports.
Authors: Angelo Michele Inchingolo, Giuseppina Malcangi, Irene Ferrara,
Assunta Patano *, Fabio Viapiano, Anna Netti, Daniela Azzollini, Anna Maria
Ciocia, Elisabetta De Ruvo, Merigrazia Campanelli, Pasquale Avantario,
Antonio Mancini, Francesco Inchingolo *, Ciro Gargiulo Isacco, Alberto
Corriero, Alessio Danilo Inchingolo, Gianna Diplama
Submitted to section: Applied Dentistry and Oral Sciences,

The objective of this systematic review was to identify and analyze the most successful and promising therapeutic alternatives available to clinicians concerning MRONJ treatment strategies

The article analyzes the subject completely by approaching all the possible aspects of the treatment. However, it requires the addition of a more recent bibliography at several levels. The two clinical cases provide relevant illustrations of the presentation.

Text-level details of possible additions of bibliographic references.

Line 20: The frequency of this pathology is low.

 A review from 2008 to 2015 suggested that the frequency of

MRONJ is small and  about 1% (range, 2–6.7%) .Dodson TB. The Frequency of Medication-related Osteonecrosis of the Jaw and its Associated Risk Factors. Oral Maxillofac Surg Clin North Am. (2015)

27:509–16. doi: 10.1016/j.coms.2015.06.003

Line 39: A more recent reference is preferable

Line 40:  Add diagnostic of MRONJ :  current or previous treatment with antiresorptive drugs,  exposed bone in the maxillofacial region that persists for more than eight weeks, no history of radiation therapy to the jaws or metastatic disease of the jaws

ref: AAOMS. Medication-Related Osteonecrosis of the Jaw-2022 Update; Rosemont Publishing & Printing Corp: Cranbury, NJ, USA, 2022.

Line 51: Capitalize the beginning of the sentence

Line 53: More recent reference is preferable

Line 57: The illustrations could be enhanced by using adapted software rather than manual coloring. With reference..

Line 86 : this illustration doesn't add much

Line 134: MBRONJ ? or MRONJ.

Line150 : MRNRJ ?

For Otto (2016) : write in minuscule " in"

For Ahrenbog (2020) : write in minuscule " in"

Line 311: Lacking a recent study that discusses the concentration of antibiotics in both necrotic and vital bone of patients with ONJ provides valuable insight into the benefits of antibiotic therapy in these patients.Ref :  Bone Concentration of Ampicillin/Sulbactam: A Pilot Study in Patients with Osteonecrosis of the Jaw .Anton Straub, et all. Int J Environ Res Public Health. 2022 Nov.

Line 319: Add recent ref. Pain characteristics in medication-related osteonecrosis of the jaws.

Haviv Y, Geller Z, Mazor S, Sharav Y, Keshet N, Zadik Y. Support Care Cancer. 2021 Feb;29(2):1073-1080. doi: 10.1007/s00520-020-05600-z. Epub 2020 Jun 29. PMID: 32601851

Line 339: More  recent ref concerning animal.

   Management of medication-related osteonecrosis of jaw: Comparison between icariin and teriparatide in a rat model.

Liu J, Mattheos N, Deng C, Su C, Wang Z, Luo N, Tang H. J Periodontol. 2021 Jan;92(1):149-158. doi: 10.1002/JPER.19-0620. Epub 2020 Jun 10. PMID: 32281098

Line 381: Add ref. Low-level laser therapy prevents medication-related osteonecrosis of the jaw-like lesions via IL-1RA-mediated primary gingival wound healing. Zheng Y, Dong X, Chen S, He Y, An J, Liu M, He L, Zhang Y. BMC Oral Health. 2023 Jan 10;23(1):14. doi: 10.1186/s12903-022-02678-1.

Line 410: add Treatment of Refractory Medicine Related Osteonecrosis of Jaw With Piezosurgical Debridement and Autologous Platelet Rich Fibrin: Feasibility Study. Gurav S, Dholam KP, Singh GP. J Craniofac Surg. 2022 May 1;33(3):e226-e230. doi: 10.1097/SCS.0000000000007981. Epub 2021 Jul 23. PMID: 34310422

Line 554 : add; The Therapeutic Effectiveness Using Fluorescence-Guided Surgery for MRONJ.  Huang H, Zhao N, Li Q, Qiao Q, Zhang J, Guo C, Guo Y. Biomed Res Int. 2022 Sep 17;2022:1650790. doi: 10.1155/2022/1650790. eCollection 2022. PMID: 36164452

Line 557: MBRONJ? if there is no error MBRONJ must appear in the abbreviations

Line 735: . example of comorbidity that can influence the MRONJ pathology   Effects of Type 2 Diabetes Mellitus on Osteoclast Differentiation, Activity, and Cortical Bone Formation in POSTmenopausal MRONJ Patients.  Park SM, Lee JH. J Clin Med. 2022 Apr 23;11(9):2377. doi: 10.3390/jcm11092377. PMID: 35566506 Free PMC article.
